# Defining hierarchical protein interaction networks from spectral analysis of bacterial proteomes

**Mark A Zaydman**[1]*[†], **Alexander S Little**[2], **Fidel Haro**[2], **Valeryia Aksianiuk**[2], **William J Buchser**[3], **Aaron DiAntonio**[4], **Jeffrey I Gordon**[1,5], **Jeffrey Milbrandt**[3], **Arjun S Raman**[2,6,7]*[†]

[1]Department of Pathology and Immunology, Washington University School of Medicine, St Louis, United States; [2]Duchossois Family Institute, University of Chicago, Chicago, United States; [3]Department of Genetics, Washington University School of Medicine, St Louis, United States; [4]Department of Developmental Biology, Washington University School of Medicine, St Louis, United States; [5]The Edison Family Center for Genome Sciences and Systems Biology, Washington University School of Medicine, St Louis, United States; [6]Department of Pathology, University of Chicago, Chicago, Chicago, United States; [7]Center for the Physics of Evolving Systems, University of Chicago, Chicago, Chicago, United States

**\*For correspondence:**
zaydmanm@wustl.edu (MAZ);
araman@bsd.uchicago.edu (ASR)

[†]These authors contributed equally to this work

**Competing interest:** The authors declare that no competing interests exist.

**Abstract** Cellular behaviors emerge from layers of molecular interactions: proteins interact to form complexes, pathways, and phenotypes. We show that hierarchical networks of protein interactions can be defined from the statistical pattern of proteome variation measured across thousands of diverse bacteria and that these networks reflect the emergence of complex bacterial phenotypes. Our results are validated through gene-set enrichment analysis and comparison to existing experimentally derived databases. We demonstrate the biological utility of our approach by creating a model of motility in *Pseudomonas aeruginosa* and using it to identify a protein that affects pilus-mediated motility. Our method, SCALES (Spectral Correlation Analysis of Layered Evolutionary Signals), may be useful for interrogating genotype-phenotype relationships in bacteria.

## Editor's evaluation

Since the inception of comparative genomics, mining phyletic patterns has been a powerful approach for the discovery of previously unknown biological interactions. The authors use a combination of singular value decomposition of the phyletic pattern matrix and random forests classification method to uncover potential protein-protein interactions. The work illustrates the utility of such methods, which are finding increasing application in addressing various computational biological problems, such as predicting protein-protein interactions from genomic information.

## Introduction

A fundamental problem in biology is to understand how proteins interact to create a complex phenotype (*Barabási and Oltvai, 2004*; *Chuang et al., 2010*; *Hartwell et al., 1999*; *Costanzo et al., 2016*). Biochemical and genetic studies have illustrated that complex behaviors emerge from layers of protein interactions: proteins interact to form complexes, complexes interact to form pathways, and pathways interact to create phenotypes (*Papin et al., 2004*; *Ravasz, 2009*; *Nurse, 2008*). Current strategies for identifying protein-protein interactions (PPIs) span both experiment and computation. Experimental

methods are rapidly becoming more high-throughput and comprehensive (*Rajagopala et al., 2014*; *Schoenrock et al., 2017*; *Häuser et al., 2014*; *Koo et al., 2017*; *Luck et al., 2017*). Computational methods based on statistical patterns of co-occurrence or co-proximity of functionally related genes first appeared shortly after publication of whole genome sequences and are an active area of research (*Eisen, 2017*; *Pellegrini et al., 2017*; *Enright et al., 2017*; *Valencia and Pazos, 2002*). More recent efforts have advanced the state-of-the-art in computational methods by incorporating evolutionary models (phylogenomics), interaction models borrowed from statistical physics, or spectral methods borrowed from signal processing (*Nagy et al., 2020*; *Franceschini et al., 2016*; *Moi et al., 2020*; *Croce et al., 2019*; *Cong et al., 2019*; *Green et al., 2021*).

Understanding how a phenotype emerges from a collection of proteins requires the ability to relate different 'scales' of interactions, from pairwise to higher-order. While higher-order interactions can be inferred from pairwise interactions, recent experimental evidence shows that such inferences are incomplete (*Kuzmin et al., 2018*). Using bacteria as a model, here we show that both pairwise and higher-order interactions can be extracted from coevolutionary statistics to create a single multi-scale hierarchical model describing the emergence of complex phenotypes.

## Results

### Global patterns of covariation between bacterial orthologs arise from phylogeny

The power of using the kingdom bacteria as a model for evolutionary analysis is the availability of thousands of high-quality proteomes originating from diverse organisms. To broadly sample extant bacterial diversity, we downloaded all bacterial proteomes from the UniProt Reference Proteome Database (*UniProt Consortium, 2019*; downloaded May 20, 2020). Each proteome was annotated using orthologous gene groups (OGGs)—a robust and computationally tractable way of inferring orthologs—and rare OGGs present in less than 1% of proteomes were filtered (*Overbeek et al., 1999*). The resulting data matrix, $D^{OGG}$, consisted of 7047 proteomes (rows) and 10,177 OGGs (columns), where each entry is the number of times an OGG was observed in a proteome (*Figure 1A*, Materials and methods; *Huerta-Cepas et al., 2017*; *Huerta-Cepas et al., 2019*).

To explore the structure of ortholog covariation across bacteria, we analyzed $D^{OGG}$ using a technique called singular value decomposition (SVD), a generalized form of principal components analysis (PCA; *Klema and Laub, 1980*). SVD defines a spectrum of components of covariation (an 'SVD spectrum') where component 1 ($SVD_1$) explains more data variance than any other component, $SVD_2$ the second most, and so on (*Figure 1B*). We observed that bacteria sharing the same phylum clustered together on $SVD_1$ through $SVD_4$ suggesting that the most dominant patterns of bacterial variation arise from phylogeny (*Figure 1C*). These observations are consistent with prior reports that strong phylogenetic signals confound PPI inference using comparative genomics (*Schäfer and Strimmer, 2005*; *Sul et al., 2018*). Taken together, $SVD_1$ to $SVD_4$ explained only 17% of the overall data variance, motivating us to ask what information lies in the remaining data variance—more phylogenetic signal, functional signal, or noise?

### Global to local patterns of bacterial covariation progressively reveal phylogeny, pathways, and protein complexes

We analyzed the SVD spectrum for information regarding (i) phylogeny, (ii) indirect interactions between proteins reflecting biological pathways, and (iii) direct (physical) interactions between proteins reflecting protein complexes. The workflow for our analysis was as follows. SVD applied to $D^{OGG}$ output three separate matrices: bacterial projections onto left-singular vectors ($U^{OGG}$), a set of singular values, and OGG projections onto right-singular vectors ($V^{OGG}$; *Figure 2A*). We determined the statistical similarity of two bacterial proteomes across the SVD spectrum by dividing $U^{OGG}$ into five-component spectral windows and computing the correlated SVD projections ('spectral correlations') for all pairs of proteomes within a spectral window (*Figure 2B and C*). We determined the extent to which two proteins in a proteome statistically interact across the SVD spectrum by (i) approximating projections of proteins onto right-singular vectors by averaging their constituent OGG content to create $V^{protein}$, (ii) dividing $V^{protein}$ into five-component spectral windows, and (iii) computing protein-protein spectral correlations (*Figure 2—figure supplement 1*, *Figure 2D and E*). We then

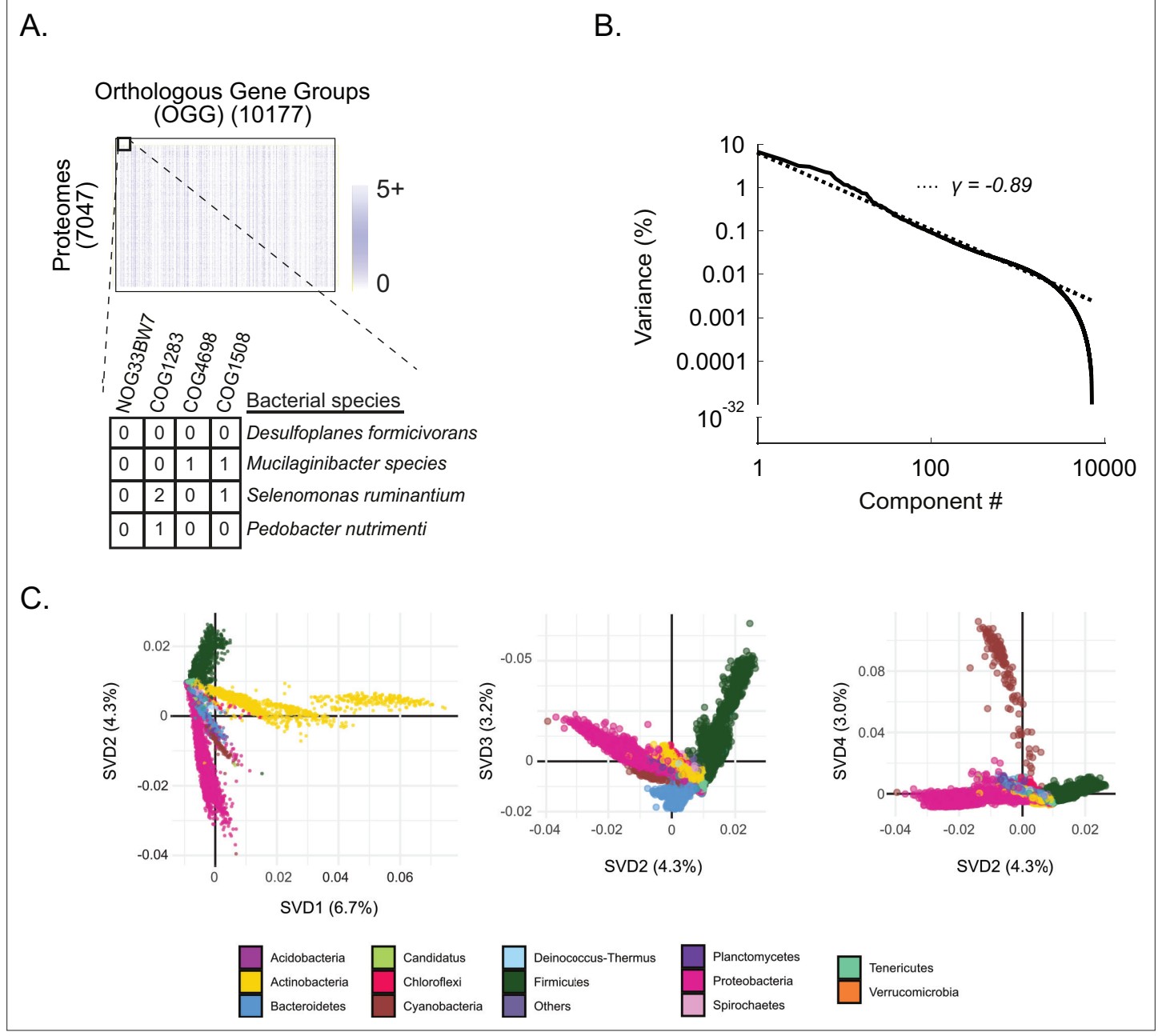

**Figure 1.** Shallow components of covariation measured across bacterial orthologs reflect broad phylogenetic relationships. (**A**) $D^{OGG}$. Rows are 7047 bacterial proteomes, columns are 10,177 orthologous gene groups (OGGs), entries are the number of annotations of an OGG within a bacterial proteome (*Figure 1—source data 1*). (**B**) Percent variance explained versus spectral component (singular value decomposition ['SVD] component') number; fit is to a power-law distribution with the indicated exponent (γ). (**C**) Contributions of bacterial proteomes (colored dots) onto SVD components 1 through 4 (percent variance indicated in parenthesis on axis labels). Dots are colored according to phylum designation (color key).

The online version of this article includes the following source data for figure 1:

**Source data 1.** $D^{OGG}$ matrix shown in *Figure 1A*.

measured the information shared between spectral correlations and the three categories of biological benchmarks using mutual information (MI) and boostrap statistical support (*Figure 2F*; Materials and methods). For an estimation of the background MI attributed to finite sampling, we shuffled the SVD projections by random permutation across a row of $V^{OGG}$ or $U^{OGG}$ and computed the MI of the shuffled projections with the benchmark (*Figure 2—figure supplement 2*). This operation maintained the distribution of SVD projections while destroying spectral correlations arising from biology, thereby

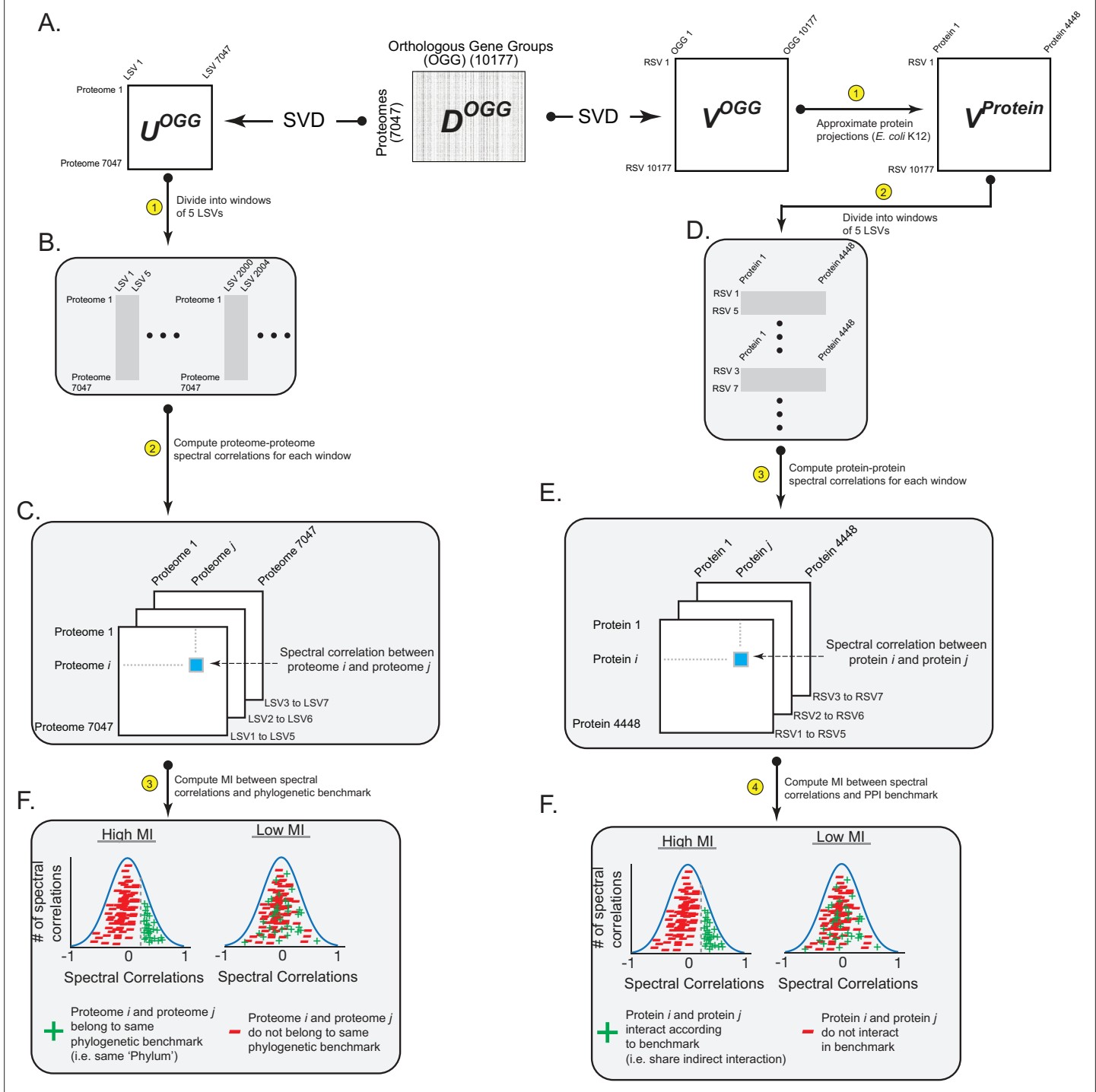

**Figure 2.** Workflow for relating patterns of ortholog covariation with phylogeny and protein interactions. (**A**) Singular value decomposition (SVD) performed on $D^{OGG}$ yields $U^{OGG}$ (rows are proteomes, columns are 'left singular vectors' [LSVs]) and $V^{OGG}$ (columns are OGGs, rows are 'right singular vectors' [RSVs]). $U^{OGG}$ is used to relate information in the SVD spectrum with phylogenetic benchmarks. $V^{OGG}$ is used to relate information in the SVD spectrum with protein interaction benchmarks. (**B,C**) Spectral correlations between bacterial proteomes are calculated by defining spectral windows of five LSVs each (panel A) and computing correlated projections between proteomes across all spectral windows (panel B). (**D,E**) Spectral correlations between proteins within a proteome are calculated by (**i**) approximating the projections of proteins onto RSVs, (ii) defining spectral windows of five RSVs each (panel C), and (iii) computing correlated projections between proteins across all spectral windows (panel D). (**F**) The final step is to compute the information shared between spectral correlations and biological benchmarks of phylogenetic relationships (left panel) or protein interactions (right panel). Shown are example distributions of spectral correlations that have 'high' and 'low' amounts of MI with a benchmark.

*Figure 2 continued on next page*

*Figure 2 continued*

The online version of this article includes the following figure supplement(s) for figure 2:

**Figure supplement 1.** Computing spectral correlations between two proteins.

**Figure supplement 2.** Computing background mutual information (MI) between spectral correlations and a benchmark.

providing a null expectation for MI arising solely from sampling constraints. Collections of biological benchmarks were created from curating publicly available databases and previously published results (Materials and methods) (*NCBI Resource Coordinators, 2018*; *Szklarczyk et al., 2019*; *Gene Ontology Consortium, 2021*; *Keseler et al., 2017*; *Cong et al., 2019*). We focused our analysis of protein interactions on the proteome of *Escherichia coli* K12 because of the wealth of existing data regarding indirect and direct PPIs for this organism.

We found that the information for the different types of benchmarks was distributed distinctly across the SVD spectrum. The top tens of components contained phylogenetic information, the top hundreds contained indirect PPI information, and the top thousands contained direct PPI information (*Figure 3A*). Even SVD components 2996 through 3000 harboring 0.025% data variance contain non-random biological information reflecting direct PPIs (*Figure 3B*). Beyond component 3000, the MI shared between protein spectral correlations and PPIs converged to the estimation of background MI. Comparing the relative distributions of these different types of information across the SVD spectrum, we observed an order of phylum, class, order, family, genus, indirect PPIs, mixed indirect/direct PPIs, and direct PPIs (*Figure 3C*, Materials and methods). The ordering of these distributions across the SVD spectrum was robust to subsampling $D^{OGG}$ to account for uneven phylogenetic distributions of bacterial strains in the UniProt database (*Figure 3—figure supplement 1*). These results show that global to local patterns of bacterial covariation reflect an intuitive hierarchy of biological scale—phylogeny, pathway, protein complex.

## A statistical approach for predicting the organization of emergent phenotype

Given the results shown in *Figure 3*, we hypothesized that by relating deep and shallow SVD components, we could create a statistical representation of emergence—the integration of local scales of protein interactions into global scales reflecting collective biological functions. Operationally, the way we related statistical information across shallow and deeper SVD components involved five steps. First, we removed SVD components enriched for phylogenetic signal and noise (components 1–33 and 3001–7047) (*Figure 4A*). Second, we computed spectral correlations between all proteins in a proteome across the remaining components (*Figure 4B*). Third, we removed spurious spectral correlations by developing a model of statistical significance. The model defined an optimal spectral window of 100 components for computing spectral correlations as well as a threshold for what constitutes 'statistically significant' spectral correlations (*Figure 4—figure supplement 1*, Materials and methods). Fourth, we defined the position in the SVD spectrum at which the spectral correlation between two proteins first dropped below the significance threshold; we term this position the 'spectral depth' of correlation between two proteins. Fifth, we discard all spectral correlations, significant or not, deeper than the spectral depth (*Figure 4C*). The effect of the last step is to condition spectral correlations found in deep regions of the SVD spectrum upon those found in shallow regions. Our rationale was that for a local interaction, like that in a protein complex, to contribute to a biological hierarchy, it must also contribute to a more global interaction network like a pathway.

Following this five-step approach, we attempted to statistically re-derive the pathway of directed motility in *E. coli* K12. We chose to study this pathway because (i) there is a wealth of previous biological data regarding its constituent parts and interactions and (ii) it is an illustrative example of a global phenotype that emerges from several layers of protein interactions. From the KEGG hierarchy of directed motility in *E. coli* K12 (KEGG hierarchy, BRITE ECO:02035), there are three levels of molecular interactions. At the lowest levels, physical interactions between proteins create small units of collective structure and function, such as a basal body, rod, ring, motor, and filament. Integration of these structures and their individual functions produces the flagellum, a machine that turns to move the cell. Integration of the flagellum and the chemotaxis system produces directed motility—the ability to move purposefully along a chemical gradient. We used the flagellar filament, FliC, as a 'bait' for

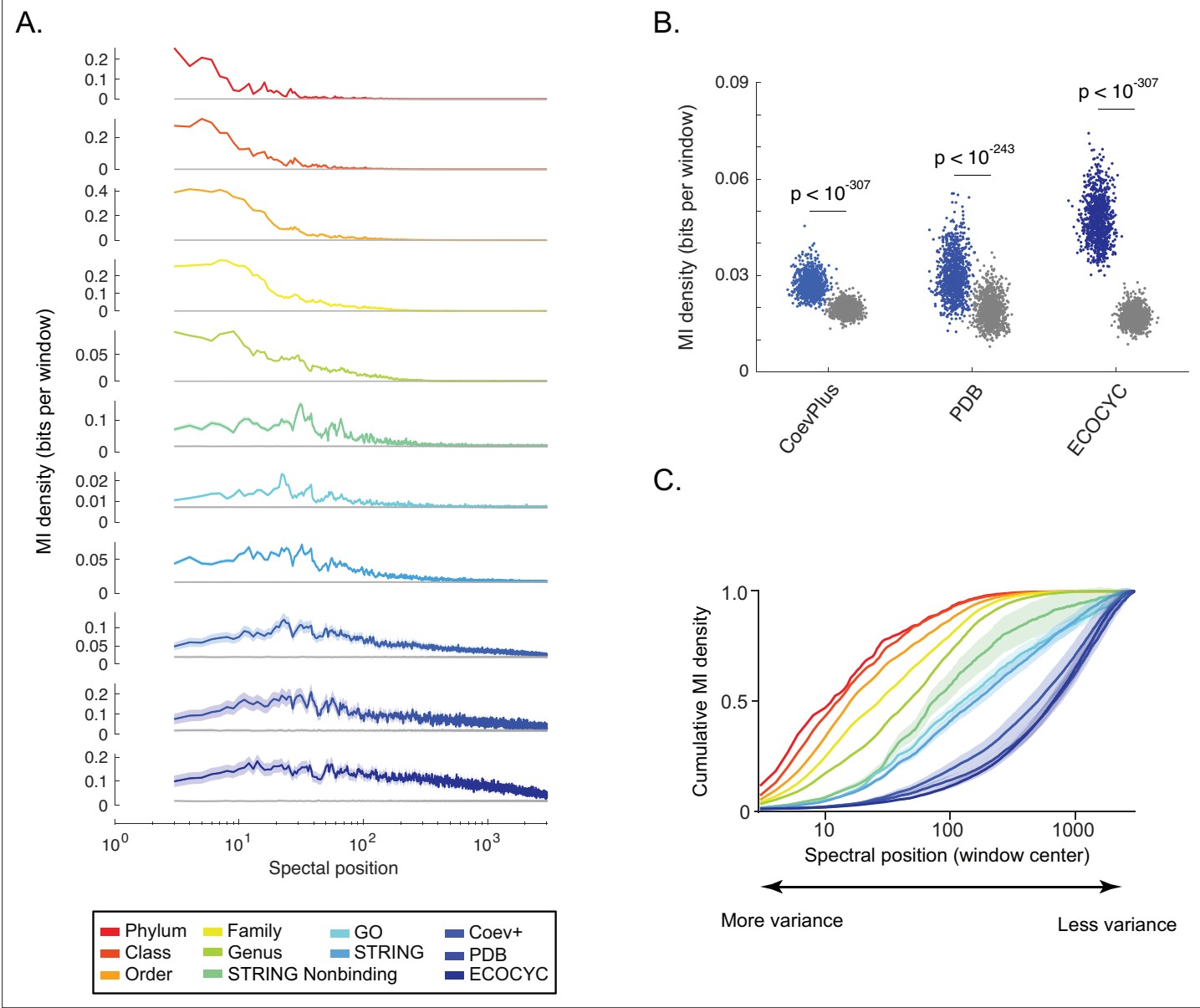

**Figure 3.** Shallow to deep spectral components of ortholog covariation reflect global to local biological 'scales'. (**A**) Distribution of information (y-axis, 'mutual information [MI] density') for each benchmark (see legend) measured across the singular value decomposition (SVD) spectrum (x-axis, 'spectral position'). Gray line in each plot is the distribution of background MI (see *Figure 2—figure supplement 2*). Lines and shaded contours represent the mean±2 standard deviations for bootstraps of each benchmark. (**B**) Information shared between three benchmarks of direct protein-protein interactions (PPIs) (x-axis) and spectral correlations computed across $SVD_{2996}$ to $SVD_{3000}$. Each dot is the MI value for a single bootstrap of the indicated benchmark. Colored dots are non-random MI, gray dots are MI values for background spectral correlations. Values of statistical significance are shown above each distribution (p-value, Student's t-test). (**C**) The degree to which information within the SVD spectrum reflects a biological benchmark, reported by the 'cumulative density of MI' (y-axis). As a curve for a benchmark approaches a value of '1', deeper spectral components contain progressively less information regarding the benchmark. Colors follow those of panel A. Shaded regions are ±2 standard deviations of the mean MI value.

The online version of this article includes the following source data and figure supplement(s) for figure 3:

**Source data 1.** NCBI taxonomic strings for each organism used to generate phylogenetic benchmarks.

**Source data 2.** Benchmarks of protein-protein interactions (PPIs) in *Escherichia coli* K12.

**Figure supplement 1.** Impact of down-sampling overrepresented phyla on results shown in *Figure 3*.

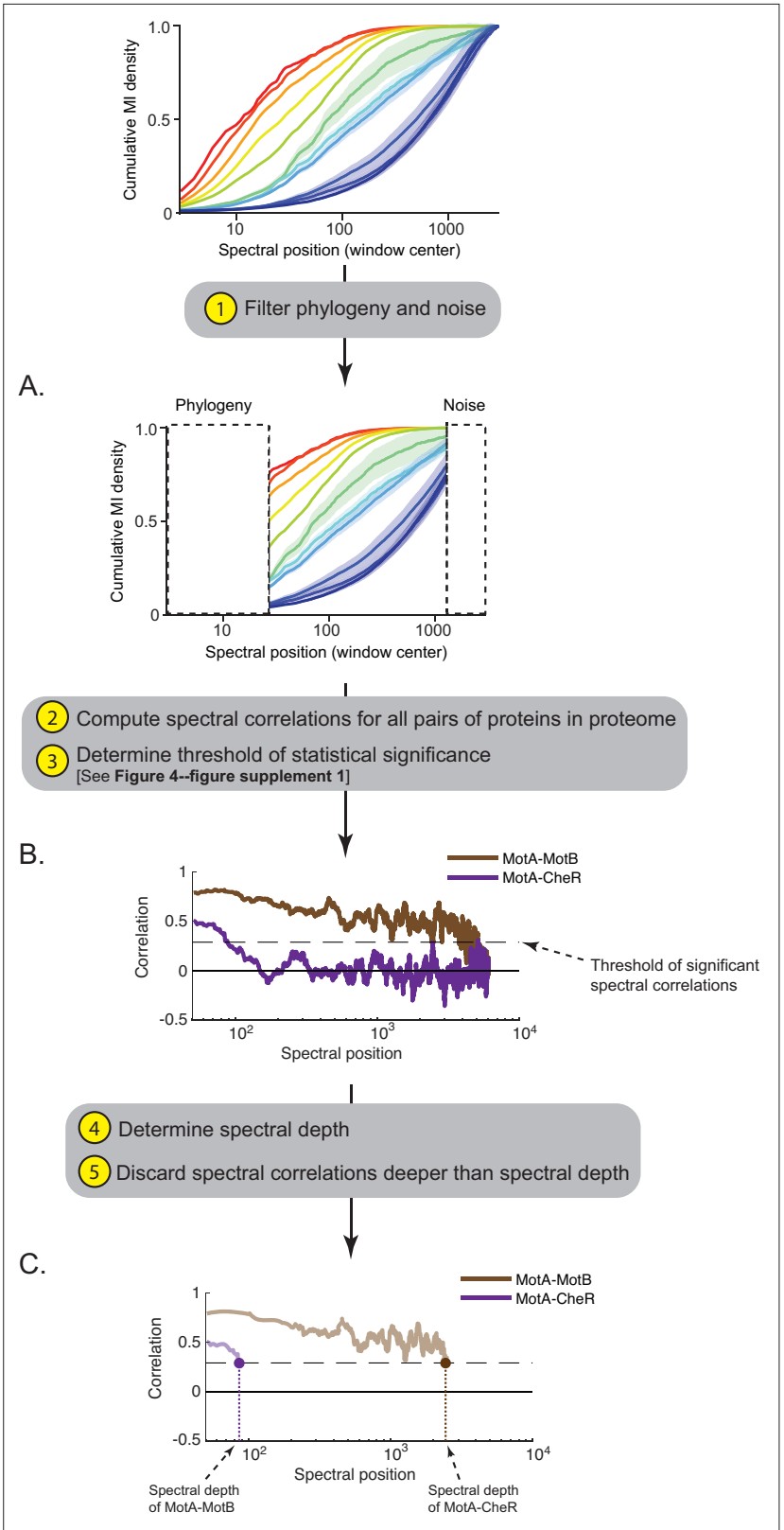

**Figure 4.** Workflow for computing the 'spectral depth' between pairs of proteins. (**A**) Spectral components enriched for indirect and direct protein interactions (25th to 75th interquartile range of cumulative mutual information [MI] density) are selected, thereby filtering components enriched for phylogeny and noise. (**B**) Spectral correlations are computed for all pairs of proteins within a proteome; spurious spectral correlations introduced by

*Figure 4 continued on next page*

*Figure 4 continued*

finite sampling are filtered. Plotted here are spectral correlations (y-axis) as a function of spectral position (x-axis) for two pairs of proteins in *Escherichia coli* K12: MotA-MotB and MotA-CheR. Dashed line reflects the threshold defining statistically significant spectral correlations. (C) 'Spectral depth' is the spectral position at which the correlation value first reaches the threshold of statistical significance. Spectral depths of MotA-MotB and MotA-CheR are shown.

The online version of this article includes the following figure supplement(s) for figure 4:

**Figure supplement 1.** Determining a threshold for statistically significant spectral correlations.

identifying spectrally correlated proteins in *E. coli* K12 putatively related to motility. Aside from this, no other biological knowledge was used: that is, no information regarding components or interactions producing complexes or collective structures contributing to directed motility. We found that 75 proteins were spectrally correlated with FliC (*Figure 5A*). We computed the spectral depth across all

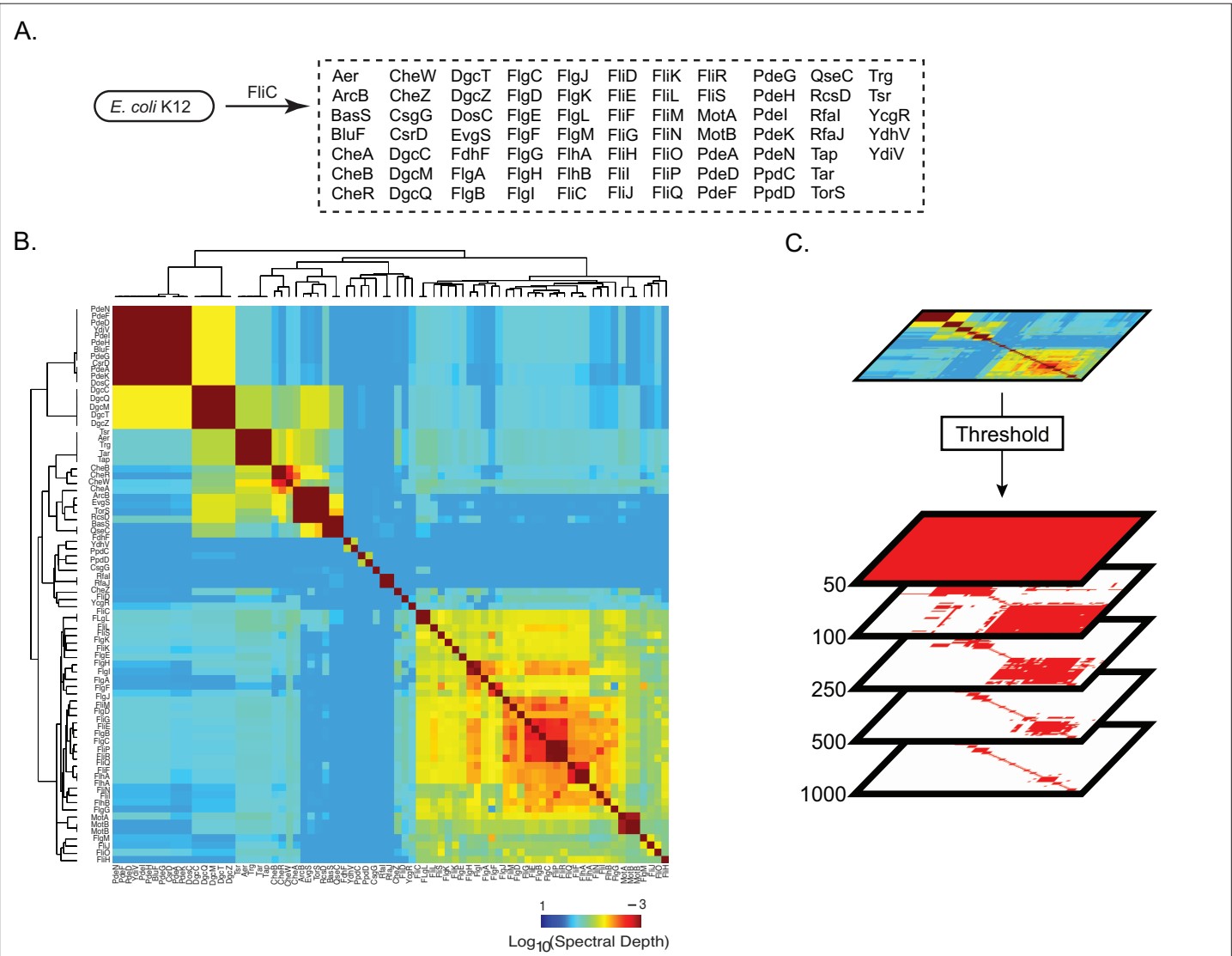

**Figure 5.** Pattern of spectral correlations with flagellar filament, FliC, in *Escherichia coli* K12. (**A**) Proteins that shared significant spectral correlations with FliC after filtering for phylogeny and noise. (**B**) Hierarchically clustered spectral depth matrix for all pairs of proteins in panel A. (**C**) Set of matrices derived from thresholding the spectral depth matrix. Red pixels indicate that two proteins have a spectral depth deeper than the indicated threshold and therefore 'statistically interact'. White pixels indicate that two proteins have a spectral depth of interaction shallower than the indicated threshold and therefore do not statistically interact.

pairs of proteins (*Figure 5B*). We observed that the pattern of spectral depths was heterogeneous: a majority of protein pairs exhibited shallow spectral depths while sparse groups of proteins exhibited deeper spectral depths (*Figure 5B*). To depict the structure of spectral depths across all pairs of proteins, we thresholded spectral depth at three different levels: 50, 300, and 1000. These thresholds were chosen to reflect areas of the SVD spectrum found to be enriched for different types of biological information per *Figure 3C*. If two proteins had a spectral depth deeper than the threshold, they were considered to 'statistically interact' (*Figure 5C*, *Supplementary file 1*). Network graphs at each thresholded spectral depth were created where nodes are proteins and edges between nodes represent a statistically significant spectral correlation between two proteins.

At a spectral depth of 50, we observed a single densely connected network devoid of obvious substructure (*Figure 6A*, top panel). Gene-set enrichment analysis (GSEA) indicated that this network was enriched for functional terms related to 'flagellar system' ($p < 10^{-45}$; *Huang et al., 2009a*; *Huang et al., 2009b*, Materials and methods). Progressing to spectral depth of 300, we observed that the network at 50 fractured into four discrete subnetworks (*Figure 6A*, middle panel). These subnetworks were significantly enriched for terms related to 'chemotaxis signaling' ($p < 10^{-15}$), 'flagellum' ($p < 10^{-56}$), 'LPS biosynthesis' ($p < 10^{-3}$), or 'cyclic di-GMP signaling' ($p < 10^{-21}$). Progressing to spectral depth of 1000, the subnetworks at 300 fractured further yielding 9 discrete subnetworks. Each subnetwork was significantly enriched for terms related to a specific function such as 'cyclic di-GMP catabolism' ($p < 10^{-25}$) and 'cyclic di-GMP synthesis' ($p < 10^{-13}$) or 'chemotransmission' ($p < 10^{-4}$) and 'chemoreception' ($p < 10^{-12}$ ; *Figure 6A*, bottom panel). Notably, considering additional spectral depths revealed further details of relevant hierarchical functional relationships within this pathway (*Figure 6—figure supplement 1*). Taken together the three network diagrams derived at spectral depths 50, 300, and 1000 depict a hierarchy of structure and function. Subnetworks observed at deeper spectral depths integrate to form the subnetworks observed at shallower spectral depths. As the subnetworks coalesced, the p-value associated with GSEA remained highly significant while the ontology of the significantly enriched terms changed. We interpret these observations to mean that as we ascend the statistical hierarchy, collective structures corresponding to new biological functions emerge from the integration of functional units at lower levels.

To validate our statistical model of motility, we compared it to the experimentally derived model of *E. coli* K12 motility detailed within the KEGG database (BR:eco02035; *Kanehisa and Goto, 2000*; *Kanehisa, 2019*; *Kanehisa et al., 2021*; *Figure 6B*). The two models were similar in several ways. First, 44 of 55 of the proteins listed in the KEGG hierarchy also appeared in the statistical hierarchy. Second, 7 of the 12 categories listed in the KEGG hierarchy had a one-to-one correspondence with a subnetwork of the statistical model sharing an overlapping set of proteins and similar descriptive label. Finally, both hierarchies shared a conserved architecture consisting of the integration of chemoreception and chemotransmission into chemotaxis signaling, the integration of flagellar substructures into the flagellum, and at the most global level the integration of chemotaxis and the flagellum. The most striking difference was that our statistical hierarchy included subnetworks related to cyclic-di-GMP signaling and LPS biosynthesis which were absent from the KEGG hierarchy. Prior experimental studies have provided direct genetic evidence that these systems are involved in *E. coli* K12 motility (*Paul et al., 2010*; *Walker et al., 2004*). Overall, of the 75 proteins in our hierarchical model of *E. coli* K12 motility, 44 (59%) were represented in the KEGG hierarchy, 28 (37%) were missing from the KEGG hierarchy but supported by prior experimental evidence in the literature, and only 3 (4%) remained unvalidated (CsgG, PpdD, TorS; *Supplementary file 1*). Taken together, these results illustrate that our approach for extracting a hierarchy of spectral correlations from the SVD spectrum produced a valid multi-scale, hierarchical model of directed motility in *E. coli* K12.

We performed four additional analyses to test the robustness and generalizability of our approach. First, we produced a hierarchical model of motility in *E. coli* K12 using MotB, the flagellar motor protein, as a query. We found a similar architecture as observed using FliC as the query with chemotaxis signaling, flagellum, and cyclic-di-GMP signaling modules appearing at spectral depth 300, and more fine-grained subnetworks appearing in deeper layers (*Figure 6—figure supplement 2Supplementary file 2*). To test generalizability across different bacteria, we created a model of motility in *Bacillus subtilis* 168 using its flagellar filament protein as a query (Hag). This analysis again produced a hierarchical model of motility that (i) recapitulated the corresponding KEGG hierarchy, (ii) identified proteins missing from the KEGG hierarchy that are known effectors of *B. subtilis*

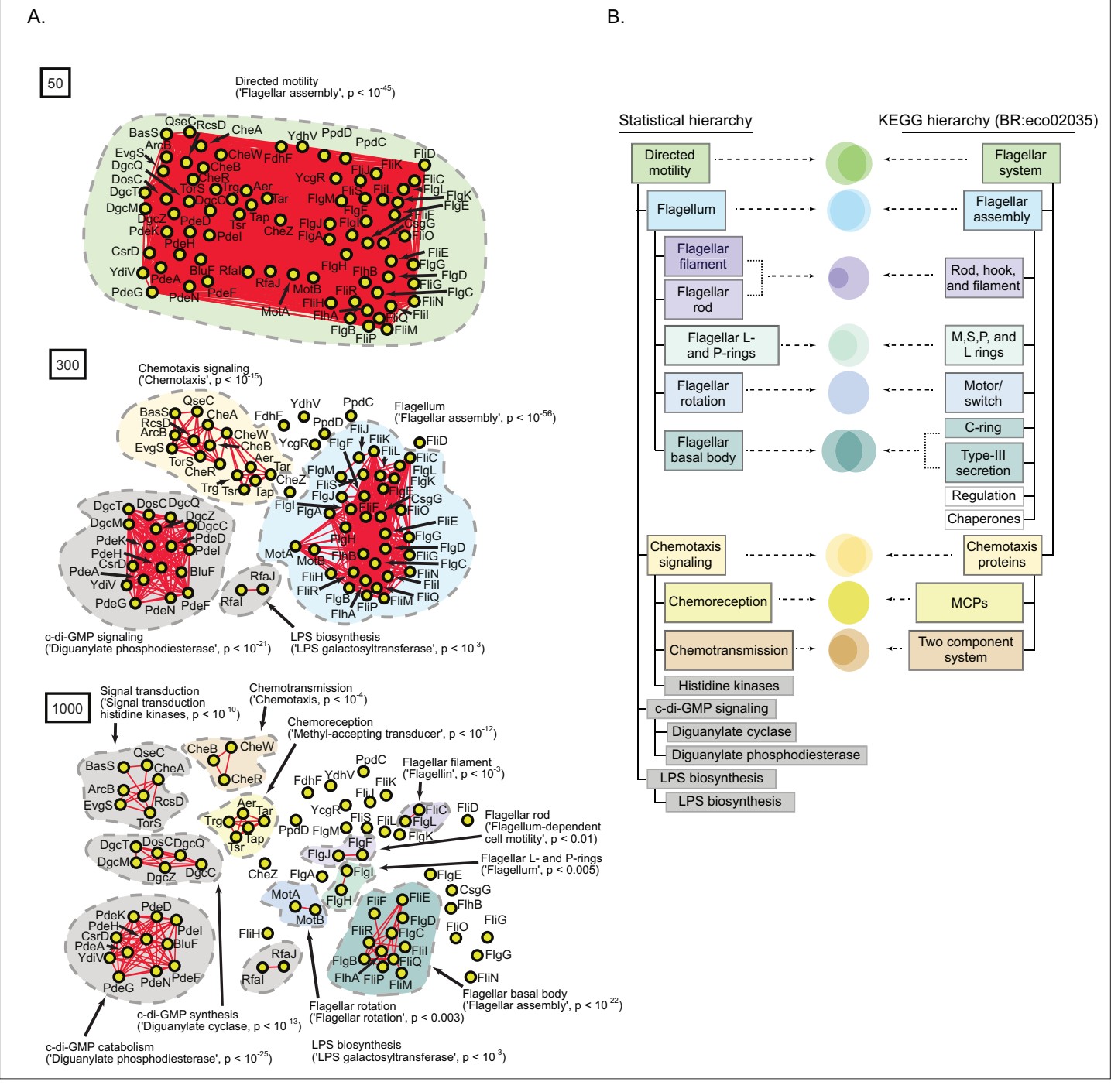

**Figure 6.** A statistically derived hierarchical model of *Escherichia coli* K12 motility. (**A**) Statistical interaction networks defined at spectral depths 50 (top), 300 (middle), and 1000 (bottom). Nodes (yellow circles) are proteins; edges (red lines) reflect statistical interactions between proteins; contours are drawn around groups of connected proteins; assignment of function associated with contours is based on gene-set enrichment analysis (GSEA) with p-value of most-enriched GSEA term in parenthesis (*Supplementary file 1*). (**B**) Comparison of the statistically derived model (left) to the KEGG model (BR:eco02035, right) of *E. coli* K12 motility. Venn diagrams represent the overlap between the sets of proteins in the indicated subnetwork of panel A (left) and the indicated KEGG category (right).

The online version of this article includes the following figure supplement(s) for figure 6:

**Figure supplement 1.** Protein interaction networks of spectrally correlated proteins with FliC in *Escherichia coli* K12 at spectral depths of 225, 500, and 750.

*Figure 6 continued on next page*

motility, and (iii) identified a small number of putative motility effectors (*Figure 6—figure supplement 3Supplementary file 3*). Next, we tested if our method could generalize to non-physically coupled pathways. We produced a model of amino acid metabolism in *E. coli* K12 using the query protein HisG, an enzyme involved in histidine biosynthesis. The resultant hierarchical model identified 130 proteins that were densely connected at spectral depth of 50. Progressing to deeper spectral depths revealed modules corresponding to specific functions, such as amino acid and nucleotide biosynthesis. At yet deeper spectral depths, modules enriched for proteins involved in the synthesis of specific amino acids became evident (*Figure 6—figure supplement 4Supplementary file 4*). Taken together, these analyses demonstrated that our approach of defining a hierarchy of spectral correlations produced valid hierarchical models of biological pathways across different query proteins, organisms, and pathways.

## Inferring functions for an uncharacterized protein from a model of emergent organization

We hypothesized that our statistical models could be used to assign general and specific functions to previously uncharacterized proteins in bacteria. To test this idea, we applied our approach to *Pseudomonas aeruginosa*, an organism with many uncharacterized proteins. We defined a hierarchical protein interaction network using PilA, a core structural element of the pilus, as a query; 141 proteins were spectrally correlated with PilA. At a spectral depth of 50, these proteins were found to collectively associate with motility of *P. aeruginosa* (*Figure 7—figure supplement 1ASupplementary file 5*; *Burrows, 2004*). Progressing to the spectral depth of 300, the global structure fractured into subnetworks associated with specific functions. Two major subnetworks related to specific types of motility, 'pilus motility' ($p<10^{-17}$) and 'bacterial flagellum' ($p<10^{-21}$), were evident, consistent with the ability of *P. aeruginosa* to use pilus or flagellar-based mechanisms of movement (*Kearns et al., 2004*; *Rashid and Kornberg, 2004*; *Figure 7A*, *Figure 7—figure supplement 1B*). Four uncharacterized proteins (Q9I5G6, Q9I5R2, Q9I0G2, Q9I0G1) were included in the pilus subnetwork that had not previously been associated with PilA in *P. aeruginosa* (*Figure 7BSupplementary file 6*). Our hierarchical models suggested that these proteins may contribute to the general function of directed motility by affecting the specific function of pilus-based motility but not flagellar-based motility.

To test these predictions, we assayed single-gene transposon mutants of *P. aeruginosa* (PAO1) for twitch-based motility mediated by the pilus or flagellar-based motility (Materials and methods) (*Kearns et al., 2004*; *Rashid and Kornberg, 2004*; *Little et al., 2018*). Transposon mutants of Q9I5R2, Q9I0G2, and Q9I0G1 exhibited motility that was not significantly different from the parent strain in both assays (*Supplementary file 7*). In contrast, we found that two different transposon mutants of Q9I5G6 exhibited significantly reduced pilus-based motility over 24, 48, and 72 hr compared to the parent strain (*Figure 7C*, $p<10^{-4}$ by Dunnett's multiple comparisons test). This phenotype resembled that of a knockout strain of PilA and was reversed upon trans-complementation. In contrast, flagellar-based motility of the transposon mutants in Q9I5G6 was not significantly different from that of the parent strain (*Figure 7C*—inset, $p>0.05$). These results illustrate that Q9I5G6 is a previously unappreciated effector of directed motility in *P. aeruginosa* that specifically impacts twitch-based motility. Compared to the background expectation of finding a protein that affects twitch motility (22 'pilus assembly proteins' in BRITE KO02035 out of 5564 proteins in the PAO1 proteome equating to a 0.4% background rate of association), our experimental results represent a statistically significant enrichment (25% association rate, $p<10^{-4}$ by chi-squared). Moreover, the results of our statistical approach shown in *Figure 7B* illustrate a far higher enrichment of functional association with identifying 19 effectors of twitch motility out of 22 proteins in the 'pilus motility module'. Overall, these experiments provide a proof of concept of how our hierarchical models may aid in discovering novel genotype-phenotype relationships.

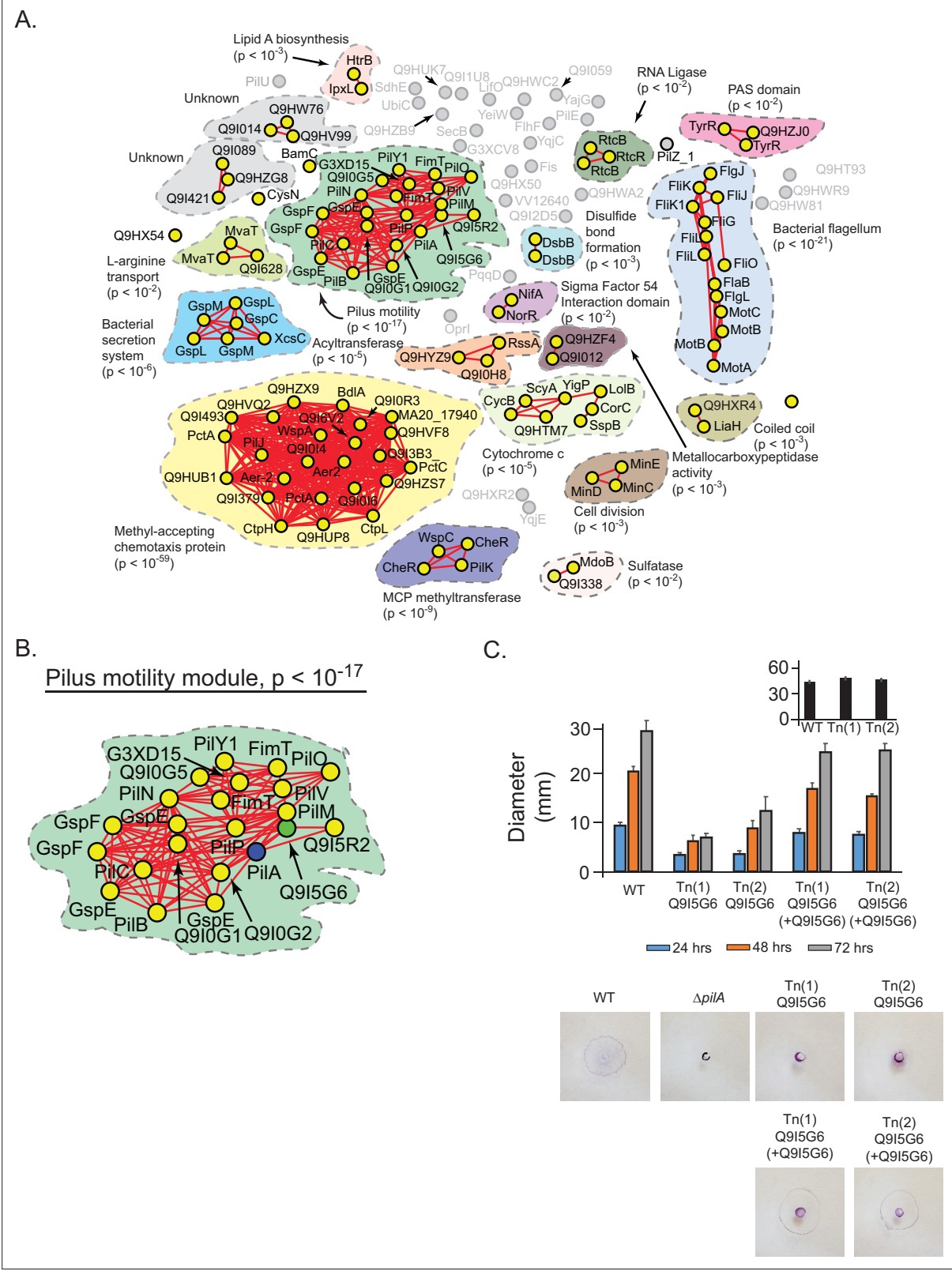

**Figure 7.** Prediction and validation of a novel effector of twitch motility in *Pseudomonas aeruginosa*. (**A**) Statistical network derived by applying a spectral depth threshold of 300 to the set of 141 protein in *P. aeruginosa* (strain PAO1) that were significantly correlated with PilA across $SVD_{34}$ to $SVD_{134}$ (see *Figure 7—figure supplement 1*). Nodes, edges, shaded contours, gene-set enrichment analysis (GSEA) enrichment terms, and p-values are defined in the same manner as in *Figure 6A*. (**B**) The pilus motility subnetwork from panel A. Nodes representing PilA and Q9I5G6 are colored blue

*Figure 7 continued on next page*

*Figure 7 continued*

and green, respectively. (**C**) Time-course of pilus-based motility for parent (WT), two transposon mutants of Q9I5G6 (Tn(1) Q9I5G6, Tn(2) Q9I5G6), and transposon mutants complemented with Q9I5G6 (Tn(1) Q9I5G6+Q9I5G6, Tn(2) Q9I5G6+Q9I5G6). Inset shows results of flagellar motility for the parent strain (WT), and the two transposon mutants of Q9I5G6 (Tn(1), Tn(2)) 24 hr post-inoculation. Representative images of the crystal-violet stained plates are shown. SVD, singular value decomposition.

The online version of this article includes the following figure supplement(s) for figure 7:

**Figure supplement 1.** Statistically derived hierarchical model of directed motility in *Pseudomonas aeruginosa* using PilA as a query.

## Using spectral correlations to predict proteome-wide functional and physical protein interaction networks

Microbiome science has taught us that diverse bacteria affect human and environmental health. Therefore, there is a critical need to expand our knowledge of biology more broadly beyond the few well-studied model organisms. Comparative genomics has been critical to inferring functionally relevant interaction networks in newly sequenced organisms. However, these methods are limited in their ability to isolate functional and physical interaction networks from each other as well as from the admixed phylogenetic signal and noise due to finite sampling (*Schäfer and Strimmer, 2005*; *Sul et al., 2018*; *Nagy et al., 2020*). Our spectral approach offers a unique opportunity to isolate interactions arising at a specific biological scale. We hypothesized that spectral correlations could be used to accurately predict proteome-wide functional and physical interaction networks across diverse bacteria.

To test this hypothesis, we classified all possible pairs of proteins in the *E. coli* K12 proteome as functionally interacting (indirect PPI), physically interacting (direct PPI), or not-interacting (Materials and methods). These predictions were based on protein-protein spectral correlations across three different spectral windows chosen to isolate phylogenetic, indirect PPI, and direct PPI information, respectively (*Figure 8—figure supplement 1*). For comparison, we predicted the same interaction classes using quantitative features of various existing computational and experimental methods. To compare the various methods, we computed F-scores for each interaction class using a composite gold-standard benchmark as well as three individual database-wide benchmarks (*Figure 8—figure supplement 2*; *Rajagopala et al., 2014*; *Babu et al., 2014*; *Babu et al., 2018*; *Hu et al., 2009*). F-score is the harmonic mean of precision and recall; to achieve an F-score of 1, the predictions must be both accurate and complete.

We found that our predictions based on windowed spectral correlations produced significantly greater F-scores for all three interaction classes compared to 18 alternative methods in all four validation tasks (*Figure 8A*, *Figure 8—figure supplement 3*, statistical comparisons by Wilcoxon rank-sum test). Of the alternative methods tested, the most mathematically similar to ours is SVD-phy (*Franceschini et al., 2016*). SVD-phy is a PCA-based approach that also uses SVD but assumes that only the top (most global) components are useful for predicting PPIs. In contrast our approach to selecting specific SVD components is data-driven, guided by the results shown in *Figure 3C*. We found that the performance of SVD-phy depended on how many of the top components were retained. Including the most shallow SVD components yielded predictions at or below the median rank of all methods across all three interaction classes. Including the shallowest 100 SVD components yielded better indirect PPI prediction without improvement of predicting direct PPIs. Including components beyond $SVD_{100}$ improved the prediction of direct PPIs while decreasing F-scores for indirect PPIs. These results show that SVD-phy can be tuned for predicting either indirect or direct interactions, but not both simultaneously. In addition, the tuned performance on either class was inferior compared to our method of windowed spectral correlations.

Our choice of SVD windows used to compute spectral correlations was guided by MI distributions generated using known *E. coli* K12 PPIs. This observation led to the concern that our approach to PPI prediction may not generalize well to other organisms. On the other hand, the underlying spectral correlations were defined from a large ensemble of bacterial proteomes, suggesting that our approach may accurately predict PPIs across diverse bacteria. To test this idea, we predicted proteome-wide direct PPIs for 11 additional phylogenetically diverse bacteria, including one organism (*Azotobacter vinelandii*) that was not represented in $D^{OGG}$ (Materials and methods). We computed the precision and recall for our predictions of any (indirect or direct), indirect, or direct PPIs using the experimentally

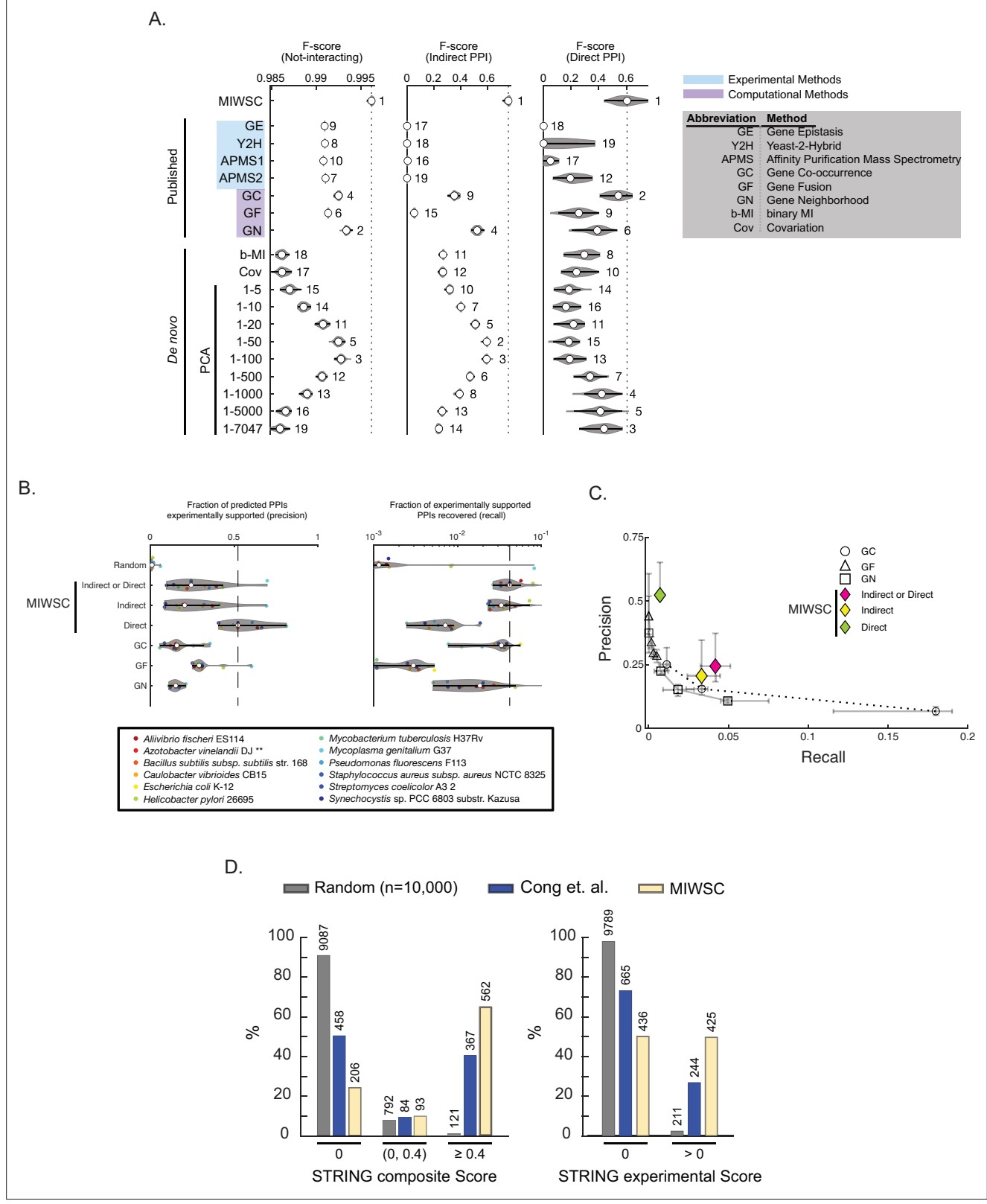

**Figure 8.** Mutual information (MI) windowed spectral correlations (MIWSCs) enable accurate classification of indirect and direct protein-protein interactions (PPIs). See *Figure 8—figure supplement 1* for definition of 'MIWSCs'. (**A**) F-scores for predicting interaction classes for *Escherichia coli* K12 protein pairs using random forest (RF) models trained on MIWSCs or features from several different methods (see legend). Violin plots show distribution of F-scores for models trained and validated on 50 random partitions of the gold-standard dataset. Numbering indicates the rank of the

*Figure 8 continued on next page*

*Figure 8 continued*

median F-score for models trained on each feature (**Figure 8—source data 1**). (**B**) Precision (left) and recall (right) for predictions of any (indirect or direct), direct, or indirect PPIs in 12 bacteria using RF models trained on the MIWSCs of *E. coli* K12 proteins benchmarked against the experimentally supported PPIs in the STRING database (experimental score > 0). Comparisons are made to a set of 10,000 randomly selected pairs and to the 'medium confidence' predictions (score > 400) in the STRING database subchannels for GC, GN, and GF. Vertical dashed line indicates the median value for the best performing method. ** in legend indicates an organism that was not part of the input dataset $D^{OGG}$ (**Figure 8—source data 1**). (**C**) Precision-recall curves were constructed for the methods of GC, GN, GF by thresholding the subchannel scores at 150 ('low confidence'), 400 ('medium confidence'), 700 ('high confidence'), and 900 ('highest confidence'). The precision versus recall is plotted for any (indirect or direct), direct, or indirect PPIs predicted using the RF models trained on MIWSCs. Symbols and whiskers represent the median and 25–75 percentile range, respectively, for the predictions produced for the 12 organisms in panel B (**Figure 8—source data 1**). (**D**) Percent of predicted direct PPIs in *Mycobacterium tuberculosis* H37Rv supported by an absent (0), low (0–0.4), or high (>0.4) composite score (left) or an absent (0) or present (>0) experimental subchannel score (right) in the STRING database. Comparisons were made between the methods of random selection (Random), amino acid coevolution (**Cong et al., 2019**), or RF models trained on MIWSC features of *E. coli* K12 proteins (MIWSC). Numbers of predicted interactions in each bin are indicated (**Figure 8—source data 1**).

The online version of this article includes the following source data and figure supplement(s) for figure 8:

**Source data 1.** Data and statistical support for random forest (RF) model validation studies, related to **Figure 8**Figure 8.

**Figure supplement 1.** Workflow for training and validating random forest (RF) models on mutual information windowed spectral correlations (MIWSCs).

**Figure supplement 2.** Workflow for training and validating various random forest (RF) models designed to predict non-interacting proteins, indirect protein-protein interactions (PPIs), and direct PPIs across the proteome of *Escherichia coli* K12.

**Figure supplement 3.** F-scores for predicting interaction classes for out-of-bag examples in the training datasets (**A**) and four additional comprehensive benchmarks (**B**).

---

supported PPIs in the STRING database as a benchmark (score > 0 in the STRING 'experimental' subchannel). We compared these results to those produced by the well-established methods of gene co-occurrence, gene neighborhood, and gene fusion by thresholding the corresponding STRING subchannel scores at different confidence levels (low, medium, high, highest). *Figure 8B* shows this comparison at the 'medium' confidence level. *Figure 8C* summarizes the entire analysis showing the precision-recall curves across all thresholds tested. We found that the sets of any indirect or direct PPIs produced by our method exhibited a higher precision for a given recall compared to the established methods. These results show that though the SVD windows used to compute spectral correlations were chosen based on analysis of PPIs in the *E. coli* K12 proteome, our approach for predicting PPIs performs as well or better than established methods across different organisms.

The state-of-the-art PPI prediction method is that produced by Cong and colleagues (Coev+) that infers direct PPIs from proteome-wide amino acid coevolution (AA Coev) (*Cong et al., 2019*). Briefly, they predict direct PPIs from primary sequence using a method called 'direct coupling analysis' (DCA) borne out of the field of statistical physics. Using the direct PPIs for *Mycobacterium tuberculosis* H37Rv found in the STRING database as a benchmark, we found that our approach exhibited significantly greater precision and recall whether using the STRING composite score (as done by Cong et al.) or the STRING experimental subscores (*Figure 8D*, Materials and methods). While we limited our comparative analysis to only *M. tuberculosis*, these results illustrate that our spectral approach to purifying biological information arising from a specific scale can produce predictions that are on par or better than a more computationally expensive alternative that leverages a higher-resolution genomic feature.

## Discussion

We developed a statistical method that predicts hierarchical protein interaction networks reflecting the emergence of complex functions in bacteria. This method involves two aspects: (i) spectral decomposition of variation across bacterial proteomes into components enriched for specific biological scales, (ii) relating covariation structure across components to define hierarchical statistical networks. These hierarchical networks closely resembled the known organization for several well-studied bacterial phenotypes. We call our approach SCALES—Spectral Correlation Analysis of Layered Evolutionary Signals. Having shown that SCALES can be useful for guiding biological discovery, we have developed the following resources: (i) a precomputed database of proteome-wide indirect (122,725,727) and direct (19,546,063) PPI predictions for all 7047 UniProt reference bacterial proteomes; (ii) a tool

for predicting indirect and direct PPIs for a user-input proteome; (iii) a tool for generating and interrogating a hierarchical model for a query protein of interest. All of these can be found at scales.cri.uchicago.edu.

## The challenge of recovering functional interactions using comparative genomics

The admixture of signals arising from phylogeny, function, and noise negatively impact the accuracy of PPI predictions using comparative genomics (*Schäfer and Strimmer, 2005*; *Sul et al., 2018*). One approach to address this problem is to use PCA which assumes that only global covariation is not statistical noise (*Wigner, 1967*; *Franceschini et al., 2016*). A known source of variability between orthologs is phylogenetic relatedness. As SVD achieves spectral clustering and phylogenetic reconstruction achieves hierarchical clustering, we do expect some level of coherence between the two approaches. However, our results also illustrate that relevant biological signal is contained throughout the SVD spectrum, including components harboring a minutiae of data variance. Components in different regions of the spectrum contained information about different biological scales—shallow components phylogeny, deeper components pathways, even deeper components protein complexes, and the very deepest noise, thereby signifying a 'cross-over' from phylogenetic to functional information. Our interpretation of these results is that while statistical variance reflecting large-scale properties can be 'compressed' into just a few shallow components, information about 'local' biological scales is distributed broadly across the spectrum. As a result, discarding global components enriched for phylogeny improved prediction of functionally relevant interactions. These results demonstrate that percent variance per component is not a good proxy for relevant biological signal. In the absence of a suitable alternative, we used knowledge in public databases to define the information content of each component. Future work will be needed to provide a theoretical basis for finding relevant signal in cases where such prior knowledge is not available.

## Extracting a multi-scale, hierarchical model of biology from coevolutionary statistics

Understanding the origins of complex biological functions requires defining hierarchical relationships describing how protein interactions integrate to create scales of biological organization. While the use of SVD is fundamental to our method, SVD itself does not define hierarchical relationships; SVD defines orthogonal components of variance ordered according to the amount of variance explained. Two results of our study were key for being able to use the SVD spectrum to produce hierarchical models. First, different components contain information regarding different biological scales. Second, the information in different components could be related by tracking the persistence of spectral correlations across components ('spectral depth'). These two results enabled extracting hierarchical statistical relationships that predict the integration of PPIs into complex structures approximating pathways and phenotypes. To our knowledge, this is a fundamentally new way of constructing hierarchical models. Instead of predicting pairwise interaction networks and then inferring higher-order networks, we infer an entire hierarchy directly from coevolutionary statistics.

## Comparison of our results with AA Coev

Recently, Cong and colleagues reported a method, AA Coev, for inferring direct PPIs from bacterial genome sequences. Their method represented a significant advance for two reasons: (i) it considered coevolution at the resolution of amino acids and (ii) it applied DCA, to entire bacterial proteomes. Like our method, AA Coev tries to extract signal related to a specific scale of biology from admixed signals of phylogeny and noise using coevolutionary statistics of genomes. Specifically, DCA is a statistical physical approach that isolates local (i.e. direct) interactions. However, we observed significant differences in quality and breadth of direct PPI inferences, scalability, and ability to reconstruct hierarchical networks.

When we compared our direct PPI predictions in *M. tuberculosis* with those of AA Coev, we predicted significantly more interactions with significantly greater precision. This was a surprising result given that our approach considers lower-information features—OGGs versus amino acid sequence-level information. One explanation may be that our predictions were derived by explicitly using correlations across three different spectral windows that provided information about three different scales

of biology: phylogeny, indirect PPIs, and direct PPIs. In contrast, AA Coev discards signal unrelated to direct PPIs. It may be the case that phylogenetic and pathway-level information provides context clues for enhancing subtle statistical signals related to direct PPIs. In other words, the confidence in assigning a putative physical interaction is increased by additional signals suggesting that the interaction also contributes to a pathway-level function among phylogenetically related bacteria. The power of our method may lie in its ability to simultaneously deconvolute yet leverage information related to these different scales. Further work is needed to better understand the differences between our method and AA Coev and to determine if the observed difference in PPI prediction is consistent across additional organisms outside of *M. tuberculosis*.

A striking difference between the two approaches is related to computational scalability. AA Coev is computationally expensive and, as a result, has been applied to just two organisms thus far. In contrast, we note that our approach can compute proteome-wide PPIs in a matter of minutes, enabling us to provide predictions for all 7047 UniProt Reference Proteomes. As identifying and characterizing different strains of bacteria is becoming increasingly important, scalable computational tools becomes a necessity.

A final difference between the two methods is how a hierarchy can be constructed. Cong et al. predicted collective units of structure, using a 'bottom-up' approach by putting together their direct PPI predictions. However, compared to the networks produced by SCALES, these networks tended to be smaller and less representative of higher-order collective structure, reinforcing results reported by Kuzmin et al. illustrating that building collective organization from pairwise networks results in incomplete hierarchical descriptions of biology. We therefore pose that though SCALES does not consider as information-rich of a feature as AA Coev, it may prove to be a useful framework to extract hierarchical relationships for connecting bacterial genotype with phenotype.

## Limitations

We observed two major limitations related to our use of OGGs as orthologous feature. First, many proteins have no annotated OGG—for example, 295 of the ~4000 *E. coli* K12 proteins (6.7%). These proteins cannot be assigned to interactions or units of function by our method. Second, many proteins share the same OGGs and appear to interact at all spectral depths. These putative interactions may or may not be related to biological function. We anticipate that ortholog annotations will continue to improve with additional bacterial genome sequences, helping to alleviate these limitations. Moreover, the use of phylogenomic methods that incorporate models of the phylogenetic history of proteins to improve ortholog definition might create superior inputs for the methods developed here (*Nagy et al., 2020*).

Another limitation of the methods developed here is that they are inherently 'mechanism-free': they leverage evolutionary constraint without knowledge of the specific pressures driving the selection of interactions. As a result, our methods identify functionally relevant interactors but cannot reveal their collective function or the detailed molecular basis of the interactions.

## The potential of generalizing SCALES to other biological systems

To what degree are the approaches developed here applicable to other biological systems? Practically, we note that the spectral properties of any given dataset will be unique. As such, re-application of these methods to a new dataset will require following the steps outlined in this work: creating a diverse ensemble, identifying relevant benchmarks, using the benchmarks to find different scales of interactions in the SVD spectrum, and enforcing the necessary statistical constraints to define hierarchical relationships. Sufficient diversity and suitable benchmarks may not currently be available for other systems as they were for bacteria. Moreover, the parameters used in this work to enable analysis—width of spectral windows, bin width for MI calculations, threshold of spectral correlation value used to define spectral depth—likely need to be derived de novo for other datasets. However, aside from these practical considerations, SCALES represents a statistical way to describe emergence—the integration of individual components into layers of collective units of function. The property of emergence spans biological systems, from proteins to ecosystems. Thus, while it remains to be tested, it may be true that SCALES is a generally useful approach to learning the hierarchical architecture of biological systems.

## Materials and methods

### Generating $D^{OGG}$

All bacterial proteomes (n=7047) in the 2020_02 release of the Uniprot Reference Proteome database were downloaded on May 20, 2020 (*UniProt Consortium, 2019*). OGGs were annotated using eggNOG-mapper V2 at the level of bacteria ('@2'; *Huerta-Cepas et al., 2017*; *Huerta-Cepas et al., 2019*). An OGG count matrix was assembled ($D^{OGG}$, *Figure 1A*) where rows were defined as proteomes, columns were defined as OGGs, and the value in each cell was the number of annotations an OGG in a proteome. The number of annotations was used to preserve as much information as possible versus the strategy of considering binary occurrence. All OGGs present in fewer than 1% of the proteomes were removed leaving 10,177 unique columns in $D^{OGG}$.

### Assembling benchmarks described in Figure 3A

The various benchmarks described within this section can be found in *Figure 3—source data 1*, *Figure 3—source data 2*.

#### Phylogeny benchmarks

NCBI phylogenetic strings were mapped to the NCBI taxonometric IDs for each of the 7047 bacteria represented in $D^{OGG}$ using taxonkit 5.0 (https://bioinf.shenwei.me/taxonkit/). Five different benchmarks were generated corresponding to pairs of proteomes that share identical phylogenetic substrings down to the level of phylum (n=5,841,696), class (n=2,460,194), order (n=807,338), family (n=434,753), or genus (n=267,794).

#### STRING Nonbinding benchmark

STRING database annotations were downloaded for the *E. coli* K12 proteome (STRING ID 511145) on July 22, 2019. A benchmark was assembled to include all protein pairs (n=14,793) with a non-zero combined STRING score that did not share a 'binding' action annotation. This benchmark was expected to be enriched for indirect PPIs.

#### GO terms benchmark

'Biological function' GO term annotations were mapped for the 4391 proteins in the *E. coli* K12 proteome through the UniProtKB API. A benchmark was assembled containing the 79,794 protein pairs that share at least one GO term annotation. This benchmark likely contained a mixture of indirect and direct PPIs.

#### STRING benchmark

STRING database annotations were downloaded for the *E. coli* K12 proteome (STRING ID 511145). A benchmark was assembled comprised of all (n=20,216) protein pairs with a non-zero combined STRING score. This benchmark included a mixture of pairs with (n=14,793) and without (n=5423) a 'binding' annotation and therefore is presumed to contain a mixture of direct and indirect PPIs.

#### ECOCYC benchmark

A previously published benchmark included 915 pairs of *E. coli* K12 proteins selected from the set of complexes in the ECOCYC database after intentionally excluding large complexes with greater than 10 proteins to enrich for directly interacting pairs of proteins (*Keseler et al., 2017*). This benchmark is assumed to primarily represent direct PPIs.

#### Coev+ benchmark

A previously published set of 1600 direct PPIs in *E. coli* K12 identified by a hybrid method combining the results of AA Coev and prior experimental data (*Cong et al., 2019*).

#### PDB benchmark

A previously published set of 809 direct PPIs in *E. coli* K12 selected by the criteria that they, or closely homologous proteins, have been observed to interact in a crystal structure in the PDB (*Cong et al., 2019*).

## Computing proteome-proteome spectral correlations

A row vector in the matrix $\boldsymbol{U^{OGG}}$ contains the projections of a single proteome onto each of the left singular vectors.

$$[\omega_i|1 > \cdots \omega_i|n > \cdots \omega_i|N >]$$

where $\omega_i$ is proteome $i$, $\omega_i \mid n >$ is the projection of $\omega_i$ onto the left singular vector $n$ $1 \leq n \leq N$, and $N$ is the total number of left singular vectors. The 'spectral correlation' is computed as the Pearson correlation between proteome $\omega_i$ and proteome $\omega_j$ within the window of left singular vectors spanning SVD components $a$ to $b$ ('spectral window'):

$$\rho_{\omega_i \omega_j}^{a:b} = corr\left(\omega_{i,[a:b]}, \omega_{j,[a:b]}\right)$$

## Computing protein-protein spectral correlations

A row vector in the matrix $\boldsymbol{V^{OGG}}$ contains the projections of a single OGG onto each of the right singular vectors:

$$[f_i|1 > \cdots f_i|m > \cdots f_i|M >]$$

where $f_i$ is the OGG in row $i$ of matrix $\boldsymbol{V^{OGG}}$, $f_i \mid m >$ is the projection of $f_i$ onto right singular vector $m$ ($1 \leq m \leq M$), and $M$ is the total number of SVD components. Each protein in a proteome may comprise up to several OGGs:

$$P_l^\omega = \{f_i, \ldots, f_j\}$$

where $P_l^\omega$ is the $l$ th protein in proteome $\omega$; $f_i$ is OGG $i$ and $f_j$ is OGG $j$. The contribution of a protein onto each right singular vector was estimated by averaging the contributions of each constituent OGG on each right singular vector. An example of this process is illustrated in *Figure 2—figure supplement 1A-F*. The 'spectral correlation' is computed as the Pearson correlation between protein $l$ and protein $m$ within the window of right singular vectors spanning SVD components $a$ to $b$ ('spectral window'):

$$\rho_{lm}^{a:b} = corr\left(P_{l,[a:b]}^\omega, P_{m,[a:b]}^\omega\right)$$

An example of this process is illustrated in *Figure 2—figure supplement 1G*.

## Computing MI between spectral correlations and benchmarks of biological knowledge

Spectral correlations were computed by first segmenting the top 3000 SVD components into windows comprised of five components each and calculating spectral correlations across all pairs of variables (either proteome-proteome or protein-protein) within each window. Computing the MI between the distribution of spectral correlations within a window and a biological benchmark quantitates how much knowing the distribution of spectral correlations reduces uncertainty about the benchmark. Because spectral correlations within a window comprise a finite distribution (i.e. not infinite), there exists intrinsic uncertainty due to the need to define a bin width in the distribution. The process of computing this uncertainty has been formalized and is calculated as the 'differential entropy' (*Cover and Thomas, 2005*):

$$H\left(\rho^{a:b}\right) = -\sum_i \Delta p\left(\rho_i^{a:b}\right) log_2 p\left(\rho_i^{a:b}\right) - log_2\left(\Delta\right)$$

$\rho^{a:b}$ is the vector of spectral correlations over the window ranging from SVD components $a$ to $b$, $H\left(\rho^{a:b}\right)$ is the differential entropy of $\rho^{a:b}$, $\rho_i^{a:b}$ is the $i$th set of correlation values, $p\left(\rho_i^{a:b}\right)$ is the probability of observing a correlation value within $\rho_i^{a:b}$, and $\Delta$ is the width of the quantization bins. In the present study $\Delta = 0.25$.

MI is the difference between the intrinsic uncertainty in spectral correlations, $H\left(\rho^{a:b}\right)$, and the uncertainty computed given a benchmark $c$, $H\left(\rho^{a:b}|c\right)$.

$$I\left(\rho^{a:\,b}, c\right) = H\left(\rho^{a:\,b}\right) - H\left(\rho^{a:\,b}|c\right)$$

The uncertainty in spectral correlations given knowledge of a benchmark is evaluated in the following way. A benchmark $c$ can be either '1' if two variables share the benchmark (i.e. two proteins share the same phylogenetic classification or two proteins are found to interact in a database) or '0' if two variables do not share the benchmark. The 'conditional entropy' of spectral correlations is the differential entropy conditioned on knowledge of the probability distribution of benchmarks. So, $H\left(\rho^{a:\,b}|c\right)$ is mathematically defined as:

$$H\left(\rho^{a:\,b}|c\right) = p\,(c = 1)\,H\left(\rho^{a:\,b}|c = 1\right) + p\,(c = 0)\,H\left(\rho^{a:\,b}|c = 0\right)$$

where $p(c=1)$ and $p(c=0)$ are the probability of observing a '1' or '0' in $c$ respectively; and $H\left(\rho^{a:\,b}|c = 1\right)$ and $H\left(\rho^{a:\,b}|c = 0\right)$ are the differential entropies of spectral correlations conditioned upon variables that share ($c=1$) and do not share ($c=0$) a benchmark, respectively. A model of random MI was generated by computing the MI shared between the spectral correlations within row-shuffled versions of $\boldsymbol{U^{OGG}}$ or $\boldsymbol{V^{OGG}}$ and the benchmarks of phylogeny and protein interactions, respectively (*Figure 2—figure supplement 2*).

For each phylogenetic benchmark, 100 bootstraps were generated consisting of equal numbers of randomly selected pairs of proteomes that do or do not share an identical phylogenetic substring annotation in the benchmark. For each protein interaction benchmark, 1000 bootstraps were generated consisting of equal numbers of randomly selected pairs of proteins that do or do not share an interaction annotation in the benchmark. For bootstraps of both phylogenetic and protein interaction benchmarks, the number of pairs sharing an annotation was equal to the number of pairs indicated for each respective benchmark in the section 'Assembling benchmarks described in *Figure 3A*'.

## Calculation of MI cumulative distribution functions shown in *Figure 3C*

Each point in the MI cumulative distribution functions (cdfs) shown in *Figure 3C* was computed as following for the window centered on component $w$ of the SV

$$cdf_w = \frac{\sum_{i=1}^{w}\left(MI_i^{data} - MI_i^{random}\right)}{\sum_{i=1}^{W}\left(MI_i^{data} - MI_i^{random}\right)}$$

where $cdf_w$ is the value of the cdf at spectral position $w$, $MI_i^{data}$ is the MI observed in window $i$, $MI_i^{random}$ is the MI produced by random correlations in window $i$, $W$ is the total number of windows, and $1 \leq w \leq W$. Because we considered five-component spectral windows within the first 3000 components, $W=2997$.

## A model of spectral correlations between non-interacting proteins

To define a model of spectral correlations between non-interacting proteins, we first considered the distribution of all pairwise spectral correlations centered on $SVD_{1000}$ for the proteins encoded in the proteome of *E. coli* K12. Our rationale was that since the vast majority of proteins do not interact, the distribution of all-by-all spectral correlations approximates correlations between proteins that do not functionally interact. The variance of this distribution decreased rapidly as the correlation window widened until a width of 100 components. This motivated our choice of computing spectral correlations over sets of 100 components (*Figure 4—figure supplement 1A, B*). We computed distributions of all-by-all spectral correlations between *E. coli* K12 proteins across windows centered on different regions of the SVD spectrum and found them to be superimposable (*Figure 4—figure supplement 1C*). Additionally, we computed such distributions for proteins from other diverse bacteria and found them to be superimposable with those derived from *E. coli* K12 (*Figure 4—figure supplement 1D*). These properties enabled defining a constant threshold for significant spectral correlations between two proteins across any 100-component SVD window. The p-value derived from the empirical cumulative distribution function of this model decreased rapidly until a threshold value of 0.29 (*Figure 4—figure supplement 1E*). Therefore, we chose the value of 0.29 associated with a p-value of 0.018 as the threshold of spectral correlations signifying functional interactions between proteins derived from any bacterial proteome within any region of the SVD spectrum.

## GSEA performed on statistical model of *E. coli* K12 motility

GSEA was performed on the sets of proteins defined by the statistical networks and subnetworks using DAVID analysis (v6.8). The ontological term with the lowest p-value is indicated for each statistical module shown in *Figure 6A*. A full list of significant ontological terms and their associated p-values for each statistical module is listed in *Supplementary file 1*.

## Assaying strains of *P. aeruginosa* for pilus and flagellar motility

All *P. aeruginosa* strains used in this study were ordered from the Manoil Lab. All strains were grown at 37°C on LB supplemented with 25 µg/ml irgasan and gentamicin (75 µg/ml) as necessary. *E. coli* XL1-Blue was maintained on LB agar plates with gentamicin (15 µg/ml) as necessary.

*P. aeruginosa* growth was at 37°C on LB supplemented with 25 µg/ml irgasan and gentamicin (75 µg/ml) as necessary. Strains were assayed for subsurface twitching motility as previously described (*Alm and Mattick, 1995*, *Little et al., 2018*). Strains were grown overnight and stab inoculated in the interstitial space between the basal surface of 1.0% LB agar and a plastic Petri dish. Plates were incubated for 48 hr at 37°C. Agar was removed and cells attached to the plate were stained with 0.5% crystal violet; twitch zone diameter was measured and plates were imaged.

Surface twitching motility assays were performed as previously described (*Little et al., 2018*; *Kearns et al., 2004*). *P. aeruginosa* strains of interest were grown overnight and concentrated in morpholine-propanesulfonic acid (MOPS) buffer (10 mM MOPS, 8 mM MgSO$_4$, pH 7.6). A 2.5 µl volume of the MOPS buffered bacterial suspension was spotted onto buffered twitching motility plates (10 mM Tris, 8 mM MgSO$_4$, 1 mM NaPO$_4$, 0.5% glucose, 1.5% agar, pH 7.6) and was incubated for 24 hr at 37°C. The twitching zone was measured and imaged.

Swimming motility was performed as previously described (*Rashid and Kornberg, 2004*). Overnight cultures were stab inoculated into the surface of LB-0.3% bacto agar and were incubated for 24 hr at 37°C. The resulting swimming zone was measured.

For complementation of genes of interest into *P. aeruginosa* strains, the complementation vector pBBR1-MCS5-PA0769 was created using Gibson assembly. The vector was transferred to *P. aeruginosa* by electroporation using 2.2 kV in a 2 mm gap cuvette and subsequent selection using gentamicin.

## Training and validating RF models for predicting PPIs in *E. coli* K12 using MIWSCs

### Assembling a 'gold-standard' dataset

Predicting PPIs is challenging in part because of the inherent class imbalance in biological systems: the number of non-interacting pairs greatly outnumber true interactors. We tried to design a relevant task by modeling this class imbalance using a previously published estimate of the ratio of these classes (*Rajagopala et al., 2014*). A 'gold-standard' dataset for *E. coli* K12 PPIs was assembled and consisted of 72,000 not-interacting, 1226 indirect PPIs, and 72 direct PPIs. All pairs defined as 'direct PPI' satisfied three criteria: they shared amino acid level coevolution (Coev+ benchmark), were annotated in the same protein complex in the ECOCYC benchmark, and interacted in the PDB benchmark. All indirect PPIs were selected based on the following criteria: they shared a 'non-binding' type interaction annotation in the STRING Nonbinding benchmark, shared a 'biological function' interaction in the GO benchmark, and did not share an interaction annotation in any of the benchmarks of direct PPIs (Coev+, ECOCYC, or PDB). The 'not-interacting' pairs did not share an interaction annotation in any of the benchmarks (GO, STRING Nonbinding, STRING, Coev+, ECOCYC, or PDB). The not-interacting set was subsampled to exceed the number of physically interacting pairs by 1000-fold (*Rajagopala et al., 2014*; *Cong et al., 2019*).

The gold-standard pairs were randomly partitioned into training (60%) and validation (40%) datasets. Fifty such random partitions were generated to assess the reproducibility of the results of the machine-learning task described below. Our rationale in partitioning the entire dataset randomly, instead of independent partitioning for each interaction class, was to produce fluctuations in the number of positives and degree of class imbalance.

### Training and validating RF models

The workflow of training and validating RF models on MIWSCs is depicted in *Figure 8—figure supplement 1*. RF models consisting of 100 decision trees were trained to classify pairs of proteins in *E. coli*

K12 as not-interacting, indirect PPIs, or direct PPIs by feeding the labeled training set examples to the TreeBagger algorithm (Matlab, v2020a). This process was repeated for each random partition of the gold-standard dataset yielding an ensemble of 50 RF models per feature. Each trained RF model was subjected to three validation tasks of classifying interaction types for unlabeled pairs of *E. coli* K12 proteins in the validation portion of the gold standard (40%). The model performance was evaluated by computing an F-score for each interaction type (not-interacting, indirect PPIs, direct PPIs), where F-score is the harmonic mean of precision and recall, precision is the ratio of the number of correctly predicted interactions within a class to the total number of predicted interactions in a class, and recall is the ratio of the number of correctly predicted interactions within a class to the total number of interactions of that class. F-scores for RF models trained on MIWSCs and used to predict the validation portion of the gold standard are shown in *Figure 8A*.

## Training and validating RF models on quantitative features of existing methods

For each feature extracted from existing methods described below, RF models were trained and validated using the identical protocol as for MIWSCs (described in the section 'Training and validating RF models' for predicting PPIs in *E. coli* K12 using MIWSCs).

### Existing experimental features

Previously published datasets derived from large-scale experimental PPI screens in *E. coli* K12 were used to generate a set of four different experimental features including: gene interaction scores from a gene epistasis screen (epistasis, $n=41,820$), sum log-likelihood scores from an affinity purification mass spectrometry screen (APMS1, $n=12,801$), protein interaction scores from an affinity purification mass spectrometry screen (APMS2, $n=291$), and binary pairs from a yeast-two hybrid screen (Y2H, $n=1766$) (*Rajagopala et al., 2014*; *Babu et al., 2014*; *Babu et al., 2018*; *Hu et al., 2009*).

### Existing computational features

Gene co-occurrence, gene fusion, and gene neighborhood subscores for *E. coli* K12 (STRING ID 511145) were extracted from the STRING database (*Szklarczyk et al., 2019*; *Rajagopala et al., 2014*; *Babu et al., 2014*; *Babu et al., 2018*; *Hu et al., 2009*). Any pairs without an interaction annotation in the STRING database were assigned a subscore of zero.

### Binary MI feature

The binary MI (b-MI) feature was modeled after the popular phylogenetic profiling method of *Pellegrini et al., 2017*. First, a binary OGG content matrix was defined as follows:

$$B^{OGG} = \begin{cases} 1, D^{OGG} > 0. \\ 0, otherwise. \end{cases}$$

where $B^{OGG}$ is the binary OGG content matrix and has the same dimensions as $D^{OGG}$.

The phylogenetic profile of an OGG across all 7047 proteomes was defined as:

$$pp_i = B_i^{OGG}$$

where $pp_i$ is the phylogenetic profile of OGG $i$. The degree to which phylogenetic profiles for a pairs of proteins in the *E. coli* K12 were similar was computed by averaging the MI between phylogenetic profiles of the OGGs encoded in the protein pair. The MI shared between two phylogenetic profiles was computed using Shannon's formulation for the MI between two discrete random variables (, ; *Cover and Thomas, 2005*).

### Covariation feature

The covariation between a pair of OGGs was described by:

$$Cov_{ij} = \frac{1}{\Omega} \sum_{\omega=1}^{\Omega} (f_i^{\omega} - <f_i>)(f_j^{\omega} - <f_i>)$$

where $Cov_{ij}$ is the covariation between OGGs $i$ and $j$, $\Omega$ is the total number of proteomes (rows) in $\boldsymbol{D^{OGG}}$, $f_i^{\omega}$ is the number of OGG $i$ in proteome $\omega$, $f_j^{\omega}$ is the number of OGG $j$ in proteome $\omega$, and $< f_i >$ is the average number of OGG $i$ across all proteomes.

### PCA-based spectral correlations features

These features were inspired by the approach of Franceschini and colleagues and the typical use of SVD to produce a low rank approximation of the initial data matrix in an effort to 'denoise' the data (*Franceschini et al., 2016*). For each pair of proteins in the *E. coli* K12 proteome spectral correlations were computed as described in the section 'Computing protein-protein spectral correlations' over windows ranging from component 1 to component $k$, where $k$=5, 10, 20, 50, 100, 500, 1000, 5000, or 7047.

F-scores for RF models trained on various experimental or computational features and used to predict the validation portion of the gold standard are shown in *Figure 8A*.

## Validating RF models in two additional validation tasks

### Training dataset task

Each decision tree within an RF model was tasked with predicting interaction classes for the out-of-bag examples from the training datasets. F-scores were computed for the consensus predictions of each model.

### Comprehensive benchmark task

Biological interactions are typically sparse: the number of not-interacting pairs of proteins vastly outnumber the number of interacting pairs. As such, we desired to challenge each of the RF models in a validation task reflective of this asymmetry. To do so, each RF model was tasked with predicting classes for all pairs of proteins in the *E. coli* K12 proteome after exclusion of pairs used in the gold-standard dataset. These predictions were validated against four different comprehensive benchmarks: the indirect PPIs in the STRING Nonbinding benchmark (*n*=5423 indirect PPIs, 9,637,213 not-interacting), the mixed indirect/direct PPIs in the GO (*n*=79,794 indirect or direct PPIs, 9,562,842 not-interacting) and STRING benchmarks (*n*=20,216 indirect or direct PPIs, 9,622,420 not-interacting), and the direct PPIs in the entire PDB benchmark (*n*=809 direct PPIs, 9,614,827 not-interacting).

F-scores for RF models trained on MIWSCs, published experimental, or published computational features and used to predict the out-of-bag examples of non-interacting proteins, indirect PPIs, or direct PPIs are shown in *Figure 8—figure supplement 3*.

## Predicting proteome-wide direct PPIs for 11 phylogenetically unrelated bacteria

### Proteomes represented in $D^{OGG}$

Each of the 50 RF models trained to classify interactions in *E. coli* K12 using MIWSCs were used to predict proteome-wide indirect and direct PPIs in the following bacteria (Uniprot Proteome ID, NCBI taxonomy ID in parentheses): *Aliivibrio fischeri* ES114 (UP000000537, 312309), *A. vinelandii* DJ (UP000002424, 322710), *B. subtilis* 168 (UP000001570, 224308), *Caulobacter vibrioides* (UP000053705, 155892), *Helicobacter pylori* 26695 (UP000000429, 85962), *M. tuberculosis* H37Rv (UP000001584, 83332), *Mycoplasma genitalium* G37 (UP000000807, 243273), *Pseudomonas fluorescens* F113 (UP000005437, 1114970), *Staphylococcus aureus* NCTC 8325 (UP000008816, 93061), *Streptomyces coelicolor* A3(2) (UP000001973, 100226), *Synechocystis* sp. PCC 6803 (UP000001425, 1111708). For each proteome, a set of consensus PPIs was defined as those for which a majority of the models (>25) produced the same classification of 'indirect PPI' or 'direct PPI'.

### Proteomes not represented in $D^{OGG}$

To predict interactions for a proteome that was not represented in $\boldsymbol{D^{OGG}}$ (e.g. *A. vinelandii* DJ, UP000002424, 322710), OGGs were mapped using EggNOG mapper V2 and MIWSCs were extracted using the OGG projections in $\boldsymbol{V^{OGG}}$ (*Huerta-Cepas et al., 2017*; *Huerta-Cepas et al., 2019*). These features were used to predict proteome-wide indirect and direct PPIs as described for the Uniprot Reference Proteomes above.

## Validating direct PPI predictions against experimental evidence in the STRING database

The predicted direct PPIs were benchmarked against the sets of interactions in the STRING database with a non-zero experimental subchannel score for *E. coli* K12 and the 11 additional organisms described above.

## A head-to-head comparison with the approach of Cong and colleagues

Cong and colleagues have provided proteome-wide PPI predictions for *E. coli* K12 and *M. tuberculosis* H37Rv (*Cong et al., 2019*). Their predictions of *E. coli* PPIs were based on AA Coev supplemented with existing knowledge ('Coev+'). In contrast, their predictions of PPIs in *M. tuberculosis* were based on AA Coev alone ('Coev'). Therefore, for a head-to-head comparison, we compared the predictions produced by our RF models trained on MIWSCs with their PPI predictions in *M. tuberculosis*. We benchmarked these interactions using two strategies. The first strategy mirrored that used by Cong and colleagues, computing the fraction of interactions assigned a STRING combined score of 0, 0–0.4, or >0.4. The second strategy used orthogonal experimental evidence by computing the fraction of interactions assigned a STRING experimental subchannel score of 0 and >0.

## Acknowledgements

We thank Robert Y Chen, Adam Bailey, Nima Mosammaparast, and Jacqueline Payton for substantial discussion regarding this manuscript. We thank Rama Ranganathan for a critical reading of the manuscript as well as in-depth discussion. We thank Dinanath Sulakhe (Center for Research Informatics [CRI], University of Chicago) for assisting in producing the web application tools described in this manuscript. We thank Sam Light, Sampriti Mukherjee, and Eric Pamer for helpful discussions regarding experiments performed.

## Additional information

### Funding
No external funding was received for this work.

### Author contributions
Mark A Zaydman, Conceptualization, Resources, Data curation, Formal analysis, Supervision, Validation, Investigation, Visualization, Methodology, Writing – original draft, Project administration, Writing – review and editing; Alexander S Little, Fidel Haro, Valeryia Aksianiuk, Investigation; William J Buchser, Manuscript edits and guidance; Aaron DiAntonio, Manuscript edits and guidance; Jeffrey I Gordon, Manuscript edits and guidance; Jeffrey Milbrandt, Manuscript edits and guidance; Arjun S Raman, Conceptualization, Resources, Data curation, Formal analysis, Supervision, Funding acquisition, Validation, Investigation, Visualization, Methodology, Writing – original draft, Project administration, Writing – review and editing

### Author ORCIDs
Mark A Zaydman http://orcid.org/0000-0002-4236-1459
Aaron DiAntonio http://orcid.org/0000-0002-7262-0968
Jeffrey I Gordon http://orcid.org/0000-0001-8304-3548
Arjun S Raman http://orcid.org/0000-0002-0070-1953

### Decision letter and Author response
Decision letter https://doi.org/10.7554/eLife.74104.sa1
Author response https://doi.org/10.7554/eLife.74104.sa2

## Additional files

### Supplementary files

- Supplementary file 1. Data pertaining to *Figure 6A*.
- Supplementary file 2. Data related to *Figure 6—figure supplement 2*.
- Supplementary file 3. Data related to *Figure 6—figure supplement 3*.
- Supplementary file 4. Data related to *Figure 6—figure supplement 4* .
- Supplementary file 5. Data related to *Figure 7A,B* and *Figure 7—figure supplement 1*.
- Supplementary file 6. Data related to gene co-occurrence, gene fusion, gene neighborhood, and co-expression data using pilA in *Pseudomonas aeruginosa* (PAO1) as a query protein.
- Supplementary file 7. Data related to *Figure 7C*.
- Transparent reporting form

### Data availability

All data relevant to this manuscript can be downloaded, in Table format, at https://www.github.com/arjunsraman/Zaydman_et_al copy archived at swh:1:rev:b2c1091aafb726d88a925ad16e-07f617a44c8cdc. All tables are available for download in .zip format. All code used for analyses contained within the manuscript can also be found within the same github repository; please refer to Readme.m and Supplemental_Code_9_23_2020.m for relevant Matlab scripts and to reproduce results.

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
