## [Editor Report]

Since the inception of comparative genomics, mining phyletic patterns has been a powerful approach for the discovery of previously unknown biological interactions. The authors use a combination of singular value decomposition of the phyletic pattern matrix and random forests classification method to uncover potential protein-protein interactions. The work illustrates the utility of such methods, which are finding increasing application in addressing various computational biological problems, such as predicting protein-protein interactions from genomic information.

---

## [Decision Letter]

**Decision letter after peer review:**

Thank you for submitting your article "Defining hierarchical protein interaction networks from spectral analysis of bacterial proteomes" for consideration by *eLife*. Your article has been reviewed by 3 peer reviewers, and the evaluation has been overseen by a Reviewing Editor and Aleksandra Walczak as the Senior Editor. The reviewers have opted to remain anonymous.

Essential revisions:

1) The basic predictive power of their MIWSC approach is demonstrated in Figure 2C. However, we were surprised by the extremely low recall shown in the right panel. A recall of 10e-5 is very low indeed, and would preclude this method from having much practical value. We went ahead and recomputed both panels in Figure 2C, based on the author's raw prediction data that we downloaded for *E. coli* from their 'scales' website (14429 predictions). We could by-and-large confirm the left panel, but for the right panel we got recall values in the 10e-02 range, which is about 1000 times higher. Is this perhaps a mislabeled axis in the plot, or a misunderstanding on our part?

2) Also related to Figure 2C: this panel demonstrates specificity and sensitivity, but only for the author's prediction class of "direct" associations. The authors, however, also predict "indirect" associations … these in fact outnumber their "direct" predictions. Is it possible to also show specificity and sensitivity for those, similar to Figure 2C, for the same set of organisms? This could be done by taking KEGG-pathway membership or GO terms as the benchmark; in our own testing we found that the overall prediction power of their "indirect MIWSC" class is less impressive – more in line with previous efforts, which might be of interest to the reader.

3) Bacteria tree is not uniformly sequenced. There is an overrepresentation of certain lineages, e.g., of gammaproteobacteria and terrabacteria (Bacillus group) in the starting matrix. This could potentially bias the quality of the correlations that are obtained in the ``mid-range' SVD components.

4) The actual biological inferences drawn for the role of the tested gene in twitching mobility might be over-interpreted. Briefly, the authors recover 4 uncharacterized proteins (Q9I5G6, Q9I5R2, Q9I0G2, Q9I0G1) as part of their T4 pilus sub-graph and infer a general function for them in the twitching mobility. They chose Q9I5G6 because it was the only one with a supposed domain of unknown function (DUF4845). However, it should be noted that Q9I5R2 also contains another such domain DUF805 along with a Zn-ribbon domain. Further, Q9I0G2 is a T2SS secretion platform protein and Q9I0G1i is the ATPase engine for the pilus. Genomic neighborhood analysis by this referee revealed that DUF4845 likely functions with the signal peptidase in secretion. Thus, given the role of the pilus in secretion and mobility, the best one could infer is a role for DUF4845 in pilus function perhaps with a greater intersection with secretion. This could even indirectly affect the mobility function which the authors' experiments are said to support. However, the authors state right in the abstract they have uncovered a twitching mobility effector. At best they could say they have uncovered a potential component that might be functionally linked to the T4 pilus which might affect secretion or twitching mobility. Indeed, the phyletic pattern of DUF4845 does not immediately suggest that all organisms with it also possess definitive twitching mobility.

5) The authors found 4 proteins of unknown function associated with pilus motility in *P. aeruginosa*. Only one shows positive experimental results. It would be nice to have some idea how general this is, i.e. how many times proteins get associated by the approach, even if they have biologically known but totally distinct functions.

6) The authors might experiment with using a matrix that removes closely related gammaproteobacterial, actinobacterial and terrabacterial lineages to see its effects on their training and predictions.

7) When the manuscript speaks of scales of protein interactions represented in the spectrum, is this visible in the singular vectors? We would imagine that large-scale properties should correspond to distributed vectors (and not only large singular values), while local scales should intuitively correspond to localised vectors.

8) It is quite evident that the first singular vectors are related to phylogeny, which is known to be a major source of variability between orthologs (at the end SVD achieves spectral clustering, and phylogeny reconstruction hierarchical clustering, which are expected to show some level of coherence). What is more interesting is the crossover from phylogenetic to functional information.

9) The MI is not really well defined, it would be hard to reproduce on the basis of what is written in the Methods section.

10) We have some doubts concerning the gold standard. If non-interacting pairs are randomly chosen, and fractions of interacting and non-interacting pairs are given in their approximate true proportions, the non-interacting set should contain about the same number of actually interacting pairs as given in the positive examples. The choice of so many random pairs to represent non-interacting pairs is therefore dangerous and potentially misleading. It remains also unclear how the extreme bias towards non-interacting pairs in the dataset is handled (most machine-learning algorithms run into problems in this case). When partitioning into training and validation sets, is this done independently for each class, or for the entire dataset? In the latter case, the number of positive examples might fluctuate a lot in the training set.

11) The comparison with Cong et al., might be potentially spectacular, but it is hard to understand it from the little paragraph. This should be explained better. One should also keep in mind that Cong et al., use an unsupervised approach as compared to the supervised in the present manuscript.

12) The notation in Materials and methods requires revision. To give an example (beyond the MI part already mentioned), n is the number of proteomes in the first paragraph, and the number of OGGs in the second. Please use consistent notations!

13) Is Figure 1S2B really showing normalized columns? They appear optically to have very different variances, even if they all should be normalized to variance 1.

14) The code should be made available.

Presentation:

1) The paper is very hard to read and to understand. It is written in a semi-technical jargon mixing spectral analysis, machine learning and information theory. Even having expertise in these fields, we had to continuously jump between the main text, the methods and the figure (including the supplementary figures – a total of 86 pages) to follow the argumentation of the paper. This style is not suitable for a journal with a broad and interdisciplinary readership. The authors should make a serious effort to clean up their presentation, such that the main messages become accessible. Currently, even for disciplinary journals in computational biology or bioinformatics, the presentation would require simplifications.

2) The general reader would benefit from an opening figure that clearly lays out the steps in the workflow starting with SVD (giving some general background) to random forest training. It can augment what is shown in the current figure 1 and one of the supplements to figure 2.

3) The abstract repeats 8 times the word "hierarchy" or "hierarchical". Avoid such excessive repetitions.

4) The introduction was a bit short in giving credit to previous efforts in this area. Yes, the founding papers from the 90s are cited, but there have been quite a number of studies in the meantime, including the use of SVD, reviewed for example here: Nagy et al., NAR 2020, https://doi.org/10.1093/nar/gkz1241 or Moi et al., PLoS computational biology 2020, 16 (7), e1007553

5) The results on motility are really nice. We suggest placing them much earlier in the paper, and do the more technical stuff with random forest etc. later. It could come directly after Figure 1C has been explained.

[Editors’ note: further revisions were suggested prior to acceptance, as described below.]

Thank you for resubmitting your work entitled "Defining hierarchical protein interaction networks from spectral analysis of bacterial proteomes" for further consideration by *eLife*. Your revised article has been evaluated by Aleksandra Walczak (Senior Editor) and a Reviewing Editor.

The manuscript has been improved but there are some remaining issues that need to be addressed, as outlined below:

*Reviewer #1 (Recommendations for the authors):*

General Summary:

While the authors have made clearly visible efforts to improve the presentation of their manuscript, the major problem remains: while the results are very interesting, the presentation of the technical details is cumbersome, and reading the paper without massive reference to the huge number of supplementary figures is hardly possible. I think another round of simplification is unavoidable.

Detailed report:

I am still convinced that this paper has very interesting results, which fully merit publication. But in particular, the initial part of the paper is hard to read. In a nutshell, the method is quite simple: (a) extract a large matrix of phylogenetic profiles (here many proteomes annotated via OGGs), (b) perform a standard SVD, and represent OGGs via their projection to the singular vectors, (c) extract proteins having correlated projections in small spectral windows. The innovative part of the paper is a clever use of these spectral windows going from the singular values/vectors describing the large-scale organisation of the data, to smaller and smaller singular values, far beyond the few typically used in PCA, showing that these deep parts of the spectrum contain finer and finer information – here going from large functional categories via pathways to PPI. It should be possible to present this in a simple way.

I would like to make a number of more specific points:

1. The reading is impossible without the massive use of supplementary figures. To give one example (out of many), the singular vectors in U^OGG^ and V^OGG^ are used in methods but introduced only in Figure 1 Supp. 1. In addition, the left and right singular values are denoted equally as |n>, which is a bit confusing (even more since as mentioned, the matrices are not defined in the text). In general, the authors should revise the paper such that the supplement gives supplementary information, and the paper can be read without having ~50 pages of supplement in your hands.

2. The method heavily relies on subjective hyperparameters and thresholds: For the number of singular values considered, for the bin size in the MI calculation, for the size of the spectral window, for the used cutoffs of the spectral depth. Together with the somewhat anecdotal results (concentration mostly on motility), this gives the impression that the biological system and parameters are hand selected to show nicely interpretable results. I sincerely hope this is not the case, but if the authors would, e.g., show the tree generated by changing spectral depth from largest to smallest values, the different levels of organisation could become more evident (in case well-separated levels exist).

3. In some cases, it is hard to understand if the results are significant. If one considers Figure 1E, even a uniform MI distribution would have a similar shape as the blue curves, since the cumulative would be linear, and represented with a log-scale of the spectral position.

4. In some places, results imposed by the analysis are presented as astonishing findings. The most evident case is the hierarchical structure when changing spectral depth. Since spectral depth is defined as the depth where correlations are for the first time non-significant, the edge set of the graphs at higher spectral depth is necessarily proper subsets of those at lower spectral depth, and thus connected components are proper subsets, too. If all statistically significant correlations in each spectral window would have been taken into account, edges between proteins might disappear at some depth, and reappear at a deeper one, making the network potentially non-hierarchical.

5. Heavy but somewhat random statements like "Understanding the molecular basis of a phenotype requires (i) defining interactions that create units of collective function at different biological scales and (ii) relating these scales to create a hierarchical model of emergent phenotype." should be avoided.

---

## [Author Response]

Essential revisions:1) The basic predictive power of their MIWSC approach is demonstrated in Figure 2C. However, we were surprised by the extremely low recall shown in the right panel. A recall of 10e-5 is very low indeed, and would preclude this method from having much practical value. We went ahead and recomputed both panels in Figure 2C, based on the author's raw prediction data that we downloaded for *E. coli* from their 'scales' website (14429 predictions). We could by-and-large confirm the left panel, but for the right panel we got recall values in the 10e-02 range, which is about 1000 times higher. Is this perhaps a mislabeled axis in the plot, or a misunderstanding on our part?

We thank the reviewer for bringing attention to this point and recalculating the recall. We indeed found an error in our plot. We have corrected this error and observe similar results as the reviewer with recall on the order of 10^-2^ (Figure 4B).

2) Also related to Figure 2C: this panel demonstrates specificity and sensitivity, but only for the author's prediction class of "direct" associations. The authors, however, also predict "indirect" associations … these in fact outnumber their "direct" predictions. Is it possible to also show specificity and sensitivity for those, similar to Figure 2C, for the same set of organisms? This could be done by taking KEGG-pathway membership or GO terms as the benchmark; in our own testing we found that the overall prediction power of their "indirect MIWSC" class is less impressive – more in line with previous efforts, which might be of interest to the reader.

We thank the reviewer for this excellent suggestion. We have now repeated the same analysis for our indirect predictions as well as the combined set of indirect and direct predictions to provide the reader with a more comprehensive understanding of the accuracy of our method. In addition, we considered that applying different confidence thresholds to the established methods might produce different results and therefore have considered various thresholds and reconstructed complete precision-recall (PR) curves for the methods of GC, GN, and GF. These analyses provide a more fair and complete evaluation of our predictions in comparison with standard prediction tools. We have added text in the manuscript to reflect these points (lines 296-308):

“We computed the precision and recall for our predictions of any (indirect or direct), indirect, or direct PPIs using the experimentally supported PPIs in the STRING database as a benchmark (score > 0 in the STRING ‘experimental’ subchannel). We compared these results to those produced by the well-established methods of gene co-occurrence, gene neighborhood, and gene fusion by thresholding the corresponding STRING subchannel scores at different confidence levels (low, medium, high, highest). Figure 4B shows this comparison at the ‘medium’ confidence level. Figure 4C summarizes the entire analysis showing the precision-recall curves across all thresholds tested. We found that the sets of any indirect or direct PPIs produced by our method exhibited a higher precision for a given recall compared to the established methods. These results show that though the SVD windows used to compute spectral correlations were chosen based on analysis of PPIs in the *E. coli* K12 proteome, our approach for predicting PPIs performs as well or better than established methods across different organisms.”

3) Bacteria tree is not uniformly sequenced. There is an overrepresentation of certain lineages, e.g., of gammaproteobacteria and terrabacteria (Bacillus group) in the starting matrix. This could potentially bias the quality of the correlations that are obtained in the ``mid-range' SVD components.

We appreciate this comment regarding the over-representation of certain clades in the original matrix. We performed a sensitivity analysis of our results by randomly downsampling the top 4 Phyla represented in the input dataset by 50%. We have included results of our analysis in the manuscript and have added a supplemental figure (Figure 1—figure supplement 5) to summarize our findings. Below is the text (lines 102-109).

“Comparing the relative distributions of these different types of information across the SVD spectrum, we observed an order of Phylum, Class, Order, Family, Genus, indirect PPIs, mixed indirect/direct PPIs, and direct PPIs. The ordering of these distributions across the SVD spectrum was robust to sub-sampling D^OGG^ to account for uneven phylogenetic distributions of bacterial strains in the input data (Figure 1—figure supplement 5). These results show that global to local patterns of bacterial covariation reflect an intuitive hierarchy of biological scale—phylogeny, pathway, protein complex (Figure 1E, Materials and methods).”

With respect to the quality of ‘mid-range’ SVD components, we did not observe a striking difference in the MI density when comparing the subsampled matrix versus the initial matrix. However, we did observe more overlap between the ‘Indirect PPI’ benchmark (STRING NB) and the ‘Mixed indirect and direct PPI’ benchmarks (GO, STRING). This difference may arise from having fewer SVD components over which information regarding the ‘Indirect PPI’ benchmark can be spread. We interpret these results to indicate that curating an optimally diverse dataset is a difficult problem that warrants future investigation.

4) The actual biological inferences drawn for the role of the tested gene in twitching mobility might be over-interpreted. Briefly, the authors recover 4 uncharacterized proteins (Q9I5G6, Q9I5R2, Q9I0G2, Q9I0G1) as part of their T4 pilus sub-graph and infer a general function for them in the twitching mobility. They chose Q9I5G6 because it was the only one with a supposed domain of unknown function (DUF4845).

We believe the text in our manuscript was unclear with respect to how these proteins were selected. We selected these proteins because they were labeled as ‘uncharacterized’ in UniProt. The modified text is found in lines 221-223:

“Four uncharacterized proteins (Q9I5G6, Q9I5R2, Q9I0G2, Q9I0G1) were identified in the pilus subnetwork that have not previously been associated with PilA in *P. aeruginosa* (Figure 3B, Supplementary File 6).”

We found that these proteins were connected to proteins that collectively reflect the global function of ‘motility’ at a spectral depth of 50. At a spectral depth of 300, these proteins shared connections with proteins that collectively reflect the specific function of ‘pilus-based motility’ but not other proteins that collectively reflect ‘flagellar motility’. We then experimentally interrogated genetic ablation of the genes encoding *all four* proteins individually to look for an effect on both pilus and flagellar motility. Our data revealed that Q9I5G6 affected pilus motility and not flagellar motility. We presented the results of our experiments for all four proteins (Figure 3C, Supplementary File 7).

However, it should be noted that Q9I5R2 also contains another such domain DUF805 along with a Zn-ribbon domain. Further, Q9I0G2 is a T2SS secretion platform protein and Q9I0G1i is the ATPase engine for the pilus. Genomic neighborhood analysis by this referee revealed that DUF4845 likely functions with the signal peptidase in secretion. Thus, given the role of the pilus in secretion and mobility, the best one could infer is a role for DUF4845 in pilus function perhaps with a greater intersection with secretion. This could even indirectly affect the mobility function which the authors' experiments are said to support. However, the authors state right in the abstract they have uncovered a twitching mobility effector. At best they could say they have uncovered a potential component that might be functionally linked to the T4 pilus which might affect secretion or twitching mobility.

We agree with the reviewer. Our approach does not infer a mechanism for Q9I5G6. We believe the key takeaway from our results is that there was a measurable effect of Tn-Q9I5G6 on pilus-based motility and not on flagellar motility. Neither our experiments nor our statistical approach were designed to address the nature (i.e. direct/physical or indirect/functional interaction) of Q9I5G6 with the pilus. We have included text in the manuscript (lines 424-427) to reflect this concept.

“Another limitation of the methods developed here is that they are inherently ‘mechanism-free’: they leverage evolutionary constraint without knowledge of the specific pressures driving the selection of interactions. As a result, our methods identify functionally relevant interactors but cannot reveal their collective function or the detailed molecular basis of the interactions.”

Indeed, the phyletic pattern of DUF4845 does not immediately suggest that all organisms with it also possess definitive twitching mobility.

We agree, merely having DUF4845 does not necessarily mean that the organism has twitching mobility. The phenotype of ‘twitching mobility’ is likely complex and dependent on the interactions between many proteins within individual organisms. Thus, the effect of Tn-Q9I5G6, conditional on the pattern of genetic interactions of strain PAO1, is to affect twitching motility. SCALES predicts an organism-specific, not domain-specific, hierarchical interaction map for Q9I5G6.

5) The authors found 4 proteins of unknown function associated with pilus motility in *P. aeruginosa*. Only one shows positive experimental results.

We appreciate the reviewer’s concern about the predictive value of our approach. It is worth noting that a ¼ rate of true positives is impressive given the background expectation. Consider that there are 5564 proteins in the *P. aeruginosa* PA01 strain, of which 25 appear in the ‘pilus-assembly module’ within the KEGG database (BRITE KO02035, ‘pilus assembly proteins’) equating to a 0.4% background rate of association. Thus, a hit rate of 25% is substantially statistically significant (p-value = 0.0001 by Chi-squared). However, we note that the hit-rate of our statistical approach is in fact much higher than 25%. We identified 22 proteins that significantly associated with Pilus-motility, 19 of which were validated by prior experiments in the literature or our own experiments (p < 0.0001 by Chi-squared) (Figure 3B). We posit that the three proteins that did not show a phenotype in our experimental conditions might in fact affect pilus-based motility but may require different assay conditions to be revealed.

We have incorporated our response into discussion within the main text found in lines 236-246:

“These results illustrate that Q9I5G6 is a previously unappreciated effector of directed motility in *P. aeruginosa* that specifically impacts twitch-based motility. Compared to the background expectation of finding a protein that affects twitch motility (22 ‘pilus assembly proteins’ in BRITE KO02035 out of 5,564 proteins in the PA01 proteome equating to a 0.4% background rate of association), our experimental results represent a statistically significant enrichment (25% association rate, p < 10^-4^ by Chi-squared). Moreover, the results of our statistical approach shown in Figure 3B illustrate a far higher enrichment of association with identifying 19 effectors of twitch motility out of 22 proteins in the ‘Pilus motility module’. As such, these experiments provide a proof of concept of how our hierarchical models may aid in discovering novel genotype-phenotype relationships.”

It would be nice to have some idea how general this is, i.e. how many times proteins get associated by the approach, even if they have biologically known but totally distinct functions.

We agree with the reviewer’s point; it would be useful to know the overall false-positive rate of our method. However, our data show that a protein can have distinct functions at a lower scale of interaction and share a collective function at higher scales. Thus, a one-to-one mapping between protein identity and its function is not possible—our results illustrate the concept that biological ‘function’ maps onto emergent networks of protein interactions not individual proteins. Nonetheless, we point to our results in Figure 2, Figure 2—supplemental figures 4, 5, and 6, Figure 5, and Figure 5—supplemental figure 1, where all of the statistical modules characterized by GSEA show a statistically significant association with a known biological function. Moreover, the validation of our proteome-wide interaction predictions across diverse organisms (Figure 4B) revealed a consistently high precision. Together, we interpret that these results provide a general sense of a low false-positive rate for our statistically connected proteins.

6) The authors might experiment with using a matrix that removes closely related gammaproteobacterial, actinobacterial and terrabacterial lineages to see its effects on their training and predictions.

We agree with the reviewer’s comment. We have addressed this in our response to Comment #3.

7) When the manuscript speaks of scales of protein interactions represented in the spectrum, is this visible in the singular vectors? We would imagine that large-scale properties should correspond to distributed vectors (and not only large singular values), while local scales should intuitively correspond to localised vectors.

We thank the reviewers for bringing up this important point. While the reviewer’s intuition was shared by the authors as well, our data illustrate a different trend. We find that large-scale properties (i.e. protein interactions that reflect broad differences in phylogeny) are highly localized to the shallow components of variation (vectors with large singular values) while local scales are distributed across the entire spectrum of vectors. This is shown in the curves reflecting the relative distributions of mutual information across the spectrum of vectors shown in Figure 1—figure supplement 4.

What is the interpretation of this data? Large-scales properties can be ‘compressed’ into just a few variables while ‘local’ properties cannot. Thus, information regarding large-scale properties is increasingly ‘filtered out’ in deeper vectors thereby enriching for more local information. Hence, one of the main results of our analysis is that the order of the spectrum of vectors corresponds to a hierarchy of protein interactions.

We have added the above interpretation of our results into the main text in lines 345-352:

“Components in different regions of the spectrum were enriched for information about different biological scales—shallow components phylogeny, deeper components pathways, even deeper components protein complexes, and the very deepest noise, thereby signifying a ‘cross-over’ from phylogenetic to functional information. Our interpretation of these results is that while statistical variance reflecting large-scale properties can be ‘compressed’ into just a few shallow components, information about ‘local’ biological scales is distributed broadly across the spectrum. As a result, discarding global components enriched for phylogeny improved prediction of functionally relevant interactions.”

8) It is quite evident that the first singular vectors are related to phylogeny, which is known to be a major source of variability between orthologs (at the end SVD achieves spectral clustering, and phylogeny reconstruction hierarchical clustering, which are expected to show some level of coherence). What is more interesting is the crossover from phylogenetic to functional information.

We are in complete agreement with the reviewer in regard to this point. We have added text in the manuscript to reflect this distinction and explicitly point out the interesting nature of the cross-over from phylogenetic information into functional information (lines 339-348):

“One approach to address this problem is to use PCA which assumes that only global covariation is not statistical noise (Wigner, 1967; Franceschini et al., 2016). A known source of variability between orthologs is phylogenetic relatedness. As SVD achieves spectral clustering and phylogenetic reconstruction hierarchical clustering, we do expect some level of coherence between the two approaches. However, our results also illustrate that relevant biological signal is contained throughout the SVD spectrum, including components harboring a minutiae of data-variance. Components in different regions of the spectrum were enriched for information about different biological scales—shallow components phylogeny, deeper components pathways, even deeper components protein complexes, and the very deepest noise, thereby signifying a ‘cross-over’ from phylogenetic to functional information.”

9) The MI is not really well defined, it would be hard to reproduce on the basis of what is written in the Methods section.

We agree that the strategy we used to compute MI is critical to explain in the manuscript and was unclear in our initial submission. To address this we have added a supplemental figure (Figure 1—figure supplement 1) that illustrates the process of computing MI between spectral correlations and biological benchmarks step-by-step; we have also rewritten the Methods section to be more readable. The new text can be found in lines 518-562.

10) We have some doubts concerning the gold standard. If non-interacting pairs are randomly chosen, and fractions of interacting and non-interacting pairs are given in their approximate true proportions, the non-interacting set should contain about the same number of actually interacting pairs as given in the positive examples. The choice of so many random pairs to represent non-interacting pairs is therefore dangerous and potentially misleading.

We agree with the reviewer’s comment that our random sampling of non-interacting pairs may lead to imperfections in the gold-standard dataset, specifically the mis-classification of true interactors as non-interactors. The strategy we took was an attempt to mimic the true interaction class imbalance noted in biological PPI studies. As we do not know the true class imbalance, we used a previous estimate provided by Rajagopala and colleagues (Rajagopala *et al.*, 2014). While we acknowledge that our gold-standard dataset is not perfect, we do not believe that it is misleading for the following reasons. First, false negatives in the gold-standard benchmark will manifest as false positives in our predictions and the imperfections of the gold-standard benchmark will only decrease model performance. Thus, the F-scores presented in Figure 4B can be interpreted as a conservative estimate. Second, the comparison of methods was performed using the same gold-standard dataset. Because the gold-standard was not selected to favor any one method, we believe the differences in relative performances are valid.

It remains also unclear how the extreme bias towards non-interacting pairs in the dataset is handled (most machine-learning algorithms run into problems in this case). When partitioning into training and validation sets, is this done independently for each class, or for the entire dataset? In the latter case, the number of positive examples might fluctuate a lot in the training set.

As the reviewer notes, class imbalance problems are challenging for many machine-learning algorithms. We purposely made a random partitioning across the whole dataset, not a partitioning independently performed for each class (i.e. stratified random sampling). Our intention was to introduce fluctuations in the number of positives across each iteration of partitioning the dataset and training models. In this way, we could ascertain the sensitivity of the performance estimates to the model class imbalance, addressing the reviewer’s concern above. In addition, we chose to report F-score as this is a valid metric for class imbalance problems. Finally, we computed F-scores for each class individually to ensure that the model accuracy is not limited to the majority class.

We agree with the reviewer that these points are critical and have included the following text in the Methods (lines 635-638 and 652-654):

“Predicting PPIs is challenging in part because of the inherent class imbalance in biological systems: the number of non-interacting pairs greatly outnumber true interactors. We tried to design a relevant task by modeling this class imbalance using a previously published estimate of the ratio of these classes (Rajagopala et al., 2014)…Our rationale in partitioning the entire dataset randomly, instead of independent partitioning for each interaction class, was to produce fluctuations in the number of positives and degree of class imbalance.”

11) The comparison with Cong et al., might be potentially spectacular, but it is hard to understand it from the little paragraph. This should be explained better. One should also keep in mind that Cong et al., use an unsupervised approach as compared to the supervised in the present manuscript.

We agree. We have added a section in the Discussion to discuss a comparison between our results and those of Cong *et al.,* The new text can be found in lines 373-412:

“Comparison of our results with AA Coev

Recently, Cong and colleagues reported a method, AA Coev, for inferring direct PPIs from bacterial genome sequences. Their method represented a significant advance for two reasons: (i) it considered co-evolution at the resolution of amino acids and (ii) it applied Direct Coupling Analysis, to entire bacterial proteomes. […] We therefore pose that though SCALES does not consider as information-rich of a feature as AA Coev, it may prove to be a useful framework to extract hierarchical relationships for connecting bacterial genotype with phenotype.”

12) The notation in Materials and methods requires revision. To give an example (beyond the MI part already mentioned), n is the number of proteomes in the first paragraph, and the number of OGGs in the second. Please use consistent notations!

We apologize for this error. We have revised the Methods section and now use consistent notations in the resubmitted manuscript. If the reviewers and referees find any more errors, please alert us and we will correct the Methods section immediately.

13) Is Figure 1S2B really showing normalized columns? They appear optically to have very different variances, even if they all should be normalized to variance 1.

We verified that this matrix is column normalized, however decided that this figure supplement was unnecessary and have removed it from the new version of the manuscript. The column-normalized matrix can be found in Figure 1-source data 1.

14) The code should be made available.

We agree. The code is publicly available and can be found at https://github.com/arjunsraman/Zaydman_et_al

This information is in lines 814-815 in the manuscript.

Presentation:1) The paper is very hard to read and to understand. It is written in a semi-technical jargon mixing spectral analysis, machine learning and information theory. Even having expertise in these fields, we had to continuously jump between the main text, the methods and the figure (including the supplementary figures – a total of 86 pages) to follow the argumentation of the paper. This style is not suitable for a journal with a broad and interdisciplinary readership. The authors should make a serious effort to clean up their presentation, such that the main messages become accessible. Currently, even for disciplinary journals in computational biology or bioinformatics, the presentation would require simplifications.

We deeply appreciate this feedback. In the resubmitted version, we have attempted to decrease jargon and focus on a single main message: extracting a hierarchy of PPIs from statistics of evolutionary variation. To this aim, we have rehauled the entire manuscript. This revision includes (i) rewriting the main text to appeal to a broader audience, (ii) reordering the main arguments of the paper, (iii) including three workflow figures (Figure 1—figure supplement 1, Figure 2—figure supplement 1, Figure 4—figure supplement 1) to explain analytical steps pictorially, and (iv) cutting material that was not salient to the main message of the paper. With respect to the latter, we have removed the entire section on domain-based analysis because in the re-written version, it felt unnecessary. We have also consolidated Supplemental Figures from the previous submission to focus on key ideas in the manuscript, decreasing the length of the manuscript from 86 pages to 60. We hope that the reviewer agrees we have taken their feedback seriously and these revisions have improved the readability and presentation of our manuscript.

2) The general reader would benefit from an opening figure that clearly lays out the steps in the workflow starting with SVD (giving some general background) to random forest training. It can augment what is shown in the current figure 1 and one of the supplements to figure 2.

We agree. We have included text describing SVD in a general way (lines 71-75):

“To explore the structure of extant bacterial variation, we analyzed D^OGG^ using a technique called Singular Value Decomposition (SVD), a generalized form of Principal Components Analysis (PCA) (Klema and Laub, 1980). SVD defines a spectrum of components of covariation (an ‘SVD spectrum’) where component 1 (SVD_1_) explains more data-variance than any other component, SVD_2_ the second most, and so on (Figure 1B).”

3) The abstract repeats 8 times the word "hierarchy" or "hierarchical". Avoid such excessive repetitions.

We have edited the abstract to remove unnecessary repetitions. The abstract (lines 22-31) reads:

“Cellular phenotypes emerge from layers of molecular interactions: proteins interact to form complexes, pathways, and phenotypes. We show that hierarchical networks of protein interactions can be extracted from the statistical pattern of proteome variation as measured across thousands of bacteria and that these networks reflect the emergence of complex bacterial phenotypes. We validate our results through gene-set enrichment analysis and comparison to existing experimentally-derived databases. We demonstrate the biological utility of our approach by creating a model of motility in *Pseudomonas aeruginosa* and using it to identify a protein that affects pilus-mediated motility. We anticipate that our method, SCALES (Spectral Correlation Analysis of Layered Evolutionary Signals), will be useful for interrogating genotype-phenotype relationships in bacteria.”

4) The introduction was a bit short in giving credit to previous efforts in this area. Yes, the founding papers from the 90s are cited, but there have been quite a number of studies in the meantime, including the use of SVD, reviewed for example here: Nagy et al., NAR 2020, https://doi.org/10.1093/nar/gkz1241 or Moi et al., PLoS computational biology 2020, 16 (7), e1007553

We thank the reviewer for this point. We have revised the manuscript to acknowledge that this is an evolving field and have included additional relevant references including those cited by the reviewer. [Lines: 35-48]

“Biochemical and genetic studies have illustrated that complex behaviors emerge from layers of protein interactions: proteins interact to form complexes, complexes interact to form pathways, and pathways interact to create phenotypes (Papin et al., 2004; Ravasz 2009; Nurse 2008). Current strategies for identifying protein-protein interactions (PPIs) span both experiment and computation. Experimental methods are rapidly becoming more high-throughput and comprehensive (Rajagopala et al., 2014; Schoenrock et al., 2017; Hauser et al., 2014; Koo et al., 2017; Luck et al., 2020). Computational methods based on statistical patterns of co-occurrence or co-proximity of functionally related genes first appeared shortly after publication of whole genome sequences (Eisen, 1998; Pellegrini et al., 1999; Enrich et al., 1999; Valencia and Pazos, 2002). More recent efforts have advanced the state-of-the-art in computational methods by incorporating evolutionary models (phylogenomics), interaction models borrowed from statistical physics, or spectral methods borrowed from signal processing (Nagy et al., 2020; Franceschini et al., 2016; Moi et al., 2020; Croce et al., 2019; Cong et al., 2019; Green et al., 2021).”

5) The results on motility are really nice. We suggest placing them much earlier in the paper, and do the more technical stuff with random forest etc. later. It could come directly after Figure 1C has been explained.

We thank the reviewer for this suggestion which we believe has dramatically improved the presentation of our results. To this end, we have reconfigured the order of the manuscript and the figures. The result of a statistically derived hierarchy of PPIs describing bacterial motility is now Figure 2. Figure 3 is the experimental validation of our approach. Figure 4 describes the results of predicting PPIs using RF models. This reconfiguration of the paper serves not only to highlight the motility results earlier on, but also emphasize the main thrust of the study.

[Editors’ note: further revisions were suggested prior to acceptance, as described below.]

Reviewer #1 (Recommendations for the authors):General Summary:While the authors have made clearly visible efforts to improve the presentation of their manuscript, the major problem remains: while the results are very interesting, the presentation of the technical details is cumbersome, and reading the paper without massive reference to the huge number of supplementary figures is hardly possible. I think another round of simplification is unavoidable.

We appreciate the reviewer’s feedback on improving the presentation of our manuscript, specifically in regard to the need of referencing many supplemental figures while navigating main figures and text. In the resubmitted version, we made several changes to ensure that (i) the main figures reflected results as well as operational processes of how we arrived at the results and (ii) the supplemental figures were not critical for understanding key messages in the manuscript. Specific changes are listed below

1. We remade Figure 1 to only show that shallow modes of variance are enriched for phylogenetic information.

2. We made a new Figure 2 to describe our statistical workflow.

3. What used to be a supplemental figure for showing that deep spectral components are enriched for physical PPIs is now Figure 3A and Figure 3B. These results are grouped together with the finding that deep to shallow spectral components collectively illustrate a progression of biological scale from phylogenetic to functional information (Figure 3C).

4. Figure 4 describes the process of computing spectral depth.

5. What used to be a supplemental figure for describing how we create hierarchical graphs of protein interaction networks is now Figure 5.

Together, the incorporation of supplemental figures from the previous version into main figures has demonstrably changed the flow of the manuscript with respect to needing reference to supplemental materials.

I would like to make a number of more specific points:1. The reading is impossible without the massive use of supplementary figures. To give one example (out of many), the singular vectors in U^OGG^ and V^OGG^ are used in methods but introduced only in Figure 1 Supp. 1. In addition, the left and right singular values are denoted equally as |n>, which is a bit confusing (even more since as mentioned, the matrices are not defined in the text). In general, the authors should revise the paper such that the supplement gives supplementary information, and the paper can be read without having ~50 pages of supplement in your hands.

We thank the reviewer for this helpful suggestion. We have reconfigured several of the figures as well as incorporated previous supplemental figures into main figures to address the reviewer’s comment and make our manuscript more accessible (see our response to ‘General Summary’ above). In addition, we have attempted to further simplify text to make transitions easier to follow. Specifically, we have introduced simpler transition to help the reader connect between section of the manuscript. Importantly, all the matrices, including the input data matrix and those resulting from SVD, are now described in the main text, please see the specific description below.

– In the section: ‘Global to ‘local’ patterns of bacterial covariation progressively reveal phylogeny, pathways, and protein complexes’, we have edited the first paragraph to read (lines 85-90):

“We analyzed the SVD spectrum for information regarding (i) phylogenetic information, (ii) indirect interactions between proteins reflecting biological pathways, and (iii) direct (physical) interactions between proteins reflecting protein complexes. The workflow for our analysis was as follows. SVD applied to D^OGG^ output three separate matrices: bacterial projections onto left-singular vectors (U^OGG^), a set of singular values, and OGG projections onto right-singular vectors (V^OGG^) (Figure 2A).”

2. The method heavily relies on subjective hyperparameters and thresholds: For the number of singular values considered, for the bin size in the MI calculation, for the size of the spectral window, for the used cutoffs of the spectral depth.

We agree with the reviewer’s assessment. As our workflow is mostly statistical, several parameters and thresholds needed to be instantiated to conduct our analysis. However, whenever possible, we derived these parameters in a data-driven manner or performed some sensitivity analysis to show the robustness of the chosen parameter. In addition, once these parameters were set, no additional parameter tuning – i.e. the same set of parameter was used for motility and non-motility systems as well analysis of the *E. coli* proteome and proteomes from organisms distantly related to *E. coli*. Regarding the reviewer’s specific concerns:

– ‘For the number of singular values considered’:

We considered the top 3000 singular values because singular values beyond this point were not significantly enriched for biological information compared to our random model. We have edited the text to reflect this point (line 116-117).

“Beyond component 3000, the MI shared between protein spectral correlations and PPIs rapidly converged to the estimation of background MI.”

– ‘for the bin size in the MI calculation’:

Estimating the information content from a sample is practically difficult because we do not have access to the complete underlying distribution. This creates a tradeoff between a high-resolution bin size, capturing all entries of the distribution, and a low resolution bin size ensuring that all bins have more than one entry. We selected the bin width of 0.25 because it seemed to provide a balance. While we have not performed a formal sensitivity analysis on this parameter choice, we point out the following: (1) we did not adjust the bin size on a case-by-case basis and (2) the quantitative value of the MI estimate is not relevant to our conclusions, but rather the relative distribution of MI across the SVD spectrum. It is therefore unlikely that the choice of bin size will have a significant impact on our main findings.

– ‘for the size of the spectral window’:

The choice for size of the spectral window was not arbitrary. Rather, we performed a comprehensive analysis to find an optimal point that balanced resolution and signal detection, as described in Figure 4—figure supplement 1A-D. Importantly, once we decided on this window size, no further adjustment was made to this parameter. We have edited the text to clarify this point (lines 134-137).

“Third, we removed spurious spectral correlations by developing a model of statistical significance. The model defined an optimal spectral window of 100 components for computing spectral correlations as well as a threshold for what constitutes ‘statistically significant’ spectral correlations (Figure 4—figure supplement 1, Materials and methods).”

– ‘for the used cutoffs of the spectral depth’:

The depicted cutoffs for spectral depth were chosen to reflect areas of the SVD spectrum we found to be enriched for different types of biological information: 50 for the center of the shallowest 100-component spectral window, 300 for a window enriched for indirect PPI information, and 1000 for a window enriched for direct PPI information. We have edited the text to clarify this point (lines 163-166).

“To depict the structure of spectral depths across all pairs of proteins, we thresholded spectral depth at three different levels: 50, 300, and 1000. These thresholds were chosen to reflect areas of the SVD spectrum found to be enriched for different types of biological information per Figure 3C.”

Together with the somewhat anecdotal results (concentration mostly on motility), this gives the impression that the biological system and parameters are hand selected to show nicely interpretable results.

In response to the reviewer’s point, we note a few important aspects:

First, our statement that biological information is spread across the SVD spectrum (Figure 3C) was based on a comprehensive interrogation of multiple different databases containing the same type of information: GO and STRING for indirect interactions, the PDB, ECOCYC and Coev+ for direct interactions. Our rationale behind using several databases was to ensure that our results were not specific to a single database or a single type of data.

Second, while we concentrated on motility to show that we could create a hierarchical protein interaction graph, we also presented several supplemental figures demonstrating that our results are robust across (i) query protein, (ii) organism, and (iii) specific pathway. In particular, Figure 6—figure supplement 4 illustrates that our approach defines a relevant hierarchical decomposition of amino acid metabolism in *E. coli* K12. Our motivation for pursuing these analyses was to ensure that our hierarchical networks were not a result of ‘overfitting’ to specifically directed motility in *E. coli* K12 using FliC as a query.

Third, we tested the predictive potential of our approach by conducting an experiment testing whether uncharacterized proteins could be characterized using our hierarchical models (Figure 7, Figure 7—figure supplement 1). These experiments were performed in *Pseudomonas aeruginosa*, an organism unrelated to *E. coli* K12, within the context of a different type of motility thereby highlighting the generality of our approach.

Finally, the last section of our paper addresses using spectral correlations to infer protein interactions across a diversity of sequenced bacteria. Using several types of analyses, we demonstrate that our approach for inferring interactions at different biological scales produces predictions that are superior to those of current methods. Our motivation behind performing this analysis was to enable interrogation of pathways and organisms that are not currently well studied. This motivation is emphasized in the first paragraph of the section ‘Using spectral correlations to predict proteome-wide functional and physical interaction networks*’* (lines 272-274):

“Microbiome science has taught us that diverse bacteria affect human and environmental health. Therefore, there is a critical need to expand our knowledge of biology more broadly beyond the few well-studied model organisms.”

We would like to reemphasize that, while there are several parameters that had to be chosen, at no point in the manuscript did we adjust our parameters to optimize the results for a specific database, organism, or pathway. In addition, we deliberately challenged our methods by predicting results across diverse databases, organisms, and pathways to ensure that our parameter choices were robust and generalizable.

I sincerely hope this is not the case, but if the authors would, e.g., show the tree generated by changing spectral depth from largest to smallest values, the different levels of organisation could become more evident (in case well-separated levels exist).

We thank the reviewer for this suggestion. We find that changing spectral depth from largest to smallest thresholds reveals a continuum of organization. To make the above point clear, we have added a new supplemental figure showing spectral depths 225, 500, and 750 for directed motility in *E. coli* K12 (Figure 6—figure supplement 1).

3. In some cases, it is hard to understand if the results are significant. If one considers Figure 1E, even a uniform MI distribution would have a similar shape as the blue curves, since the cumulative would be linear, and represented with a log-scale of the spectral position.

We thank the reviewer for this insightful question. We agree that the cumulative distribution function for a uniform MI distribution might have a similar shape. However, we clarify that the curves in Figure 1E (now Figure 3C) are the cumulative ‘specific’ MI, that is the cumulative difference between the MI produced by the data and that produced by a randomized matrix (see Methods section titled *‘*Calculation of MI cumulative distribution functions (cdfs) shown in Figure 3C'). The specific MI for a uniform distribution would be expected to be zero thereby giving a distinct CDF curve compared to those in Figure 3C. To further clarify the question of significance, we have created a new Figure 3 where panels A and B highlight the non-random nature of the MI between protein spectral correlations and whether two proteins physically interact. We have included text to clarify this point (lines 112-116):

“The top tens of components contained phylogenetic information, the top hundreds contained indirect PPI information, and the top thousands contained direct PPI information (Figure 3A). Even SVD components 2996 through 3000 harboring 0.025% data-variance contain non-random biological information reflecting direct PPIs (Figure 3B).”

4. In some places, results imposed by the analysis are presented as astonishing findings. The most evident case is the hierarchical structure when changing spectral depth. Since spectral depth is defined as the depth where correlations are for the first time non-significant, the edge set of the graphs at higher spectral depth is necessarily proper subsets of those at lower spectral depth, and thus connected components are proper subsets, too. If all statistically significant correlations in each spectral window would have been taken into account, edges between proteins might disappear at some depth, and reappear at a deeper one, making the network potentially non-hierarchical.

The reviewer brings up a very good point: our approach does not consider spectral correlations that are significant deeper than the spectral depth of correlation. Our approach to only consider the spectral depth of correlation was to enforce a hierarchy: our definition of spectral depth is predicated on the pair of proteins being significantly spectrally correlated at all windows shallower than the spectral depth. Our intention in making this choice was to see whether we could extract a hierarchy of interactions from the SVD spectrum through a defined process, not whether the SVD spectrum naturally provides a hierarchy of interactions. We highlight this point in two separate places in the text: lines (127-129) and lines (380-387):

“Given the results shown in Figure 3, we hypothesized that by relating deep and shallow SVD components, we could create a statistical representation of emergence—the integration of local scales of protein interactions into global scales reflecting collective biological functions.”

“Understanding the origins of complex biological functions requires defining hierarchical relationships describing how protein interactions integrate to create scales of biological organization. While the use of SVD is fundamental to our method, SVD itself does not define hierarchical relationships; SVD defines orthogonal components of variance ordered according to the amount of variance explained. Two results of our study were key for being able to use the SVD spectrum to produce hierarchical models. First, different components contain information regarding different biological scales. Second, the information in different components could be related by tracking the persistence of spectral correlations across components (‘spectral depth’).”

6. Heavy but somewhat random statements like "Understanding the molecular basis of a phenotype requires (i) defining interactions that create units of collective function at different biological scales and (ii) relating these scales to create a hierarchical model of emergent phenotype." should be avoided.

We thank the reviewer for this comment. We have edited the manuscript to remove several such statements, with particular care given transitions from one section to another. These changes can be viewed in the document comparing the previously submitted version with the current resubmission.